# DWM model calibration using nacelle-mounted lidar systems

Inga Reinwardt[1], Levin Schilling[1], Peter Dalhoff[1], Dirk Steudel[2], and Michael Breuer[3]

[1]Dep. Mechanical Engineering & Production, HAW Hamburg, Berliner Tor 21, D-20099 Hamburg, Germany
[2]Dep. Turbine Load Calculation, Nordex Energy GmbH, Langenhorner Chaussee 600, D-22419 Hamburg, Germany
[3]Dep. of Fluid Mechanics, Helmut-Schmidt University Hamburg, Holstenhofweg 85, D-22043 Hamburg, Germany

**Correspondence:** Inga Reinwardt (inga.reinwardt@haw-hamburg.de)

**Abstract.** Light Detection And Ranging (LiDAR) systems have gained a great importance in today's wake characteristic measurements. The aim of this measurement campaign is to track the wake meandering and in a further step to validate the wind speed deficit in the meandering frame of reference (MFR) and in the fixed frame of reference using nacelle-mounted LiDAR measurements. Additionally, a comparison of the measured and the modelled wake degradation in the MFR was
conducted. The simulations were done with two different versions of the Dynamic Wake Meandering (DWM) model. These versions differ only in the description of the quasi-steady wake deficit. Based on the findings from the LiDAR measurements, the impact of the ambient turbulence intensity on the eddy viscosity definition in the quasi-steady deficit has been investigated and, subsequently, an improved correlation function has been determined, resulting in very good conformity between the new model and the measurements.

## 1 Introduction

Wake calculation of neighbouring wind turbines is a key aspect of every wind farm development. The aim is to estimate both, energy yield of the whole wind farm and loads on single turbines, as accurately as possible. One of the main models for calculating the wake-induced turbulence in a wind farm is the so-called Frandsen model (see, for example, Frandsen (2007)). Previous measurement campaigns have shown that this model delivers conservative results for small turbine distances
(Reinwardt et al., 2018; Gerke et al., 2018). This is particularly important for onshore wind farms in densely populated areas, where a high energy output per utilized area is crucial. In such cases, the usage of a more accurate description of the physical behaviour of the wake, as defined in the DWM model, seems appropriate. The DWM model is based on the assumption that the wake behaves as a passive tracer, which means, the wake itself is deflected in vertical and horizontal direction (Larsen et al., 2008b). The combination of this deflection and the shape of the wind speed deficit leads to an increased turbulence at a
fixed position downstream. This plays an eminent role for the loads of a turbine located downstream of another turbine (Larsen et al., 2013). Therefore, a precise description of the meandering itself and the wind speed deficit in the meandering frame of reference (MFR) as well as a detailed validation of the wind speed deficit definition is fundamental.

LiDAR systems are highly suitable for wake validation purposes. Especially, the so-called scanning LiDAR systems offer great potential for detailed wake analysis. These LiDARs are capable of scanning a three-dimensional wind field, so that the
line of sight (LOS) wind speed can be measured subsequently at different positions in the wake, thus enabling the detection

of the wake meandering as well as the shape of the wind speed deficit in the MFR. That is the reason why such a device is used in the measurement campaign outlined here. Several different measurement campaigns with ground based and nacelle-mounted LiDAR systems have already been carried out in the last years, some of them even with the purpose of tracking wake meandering and validation of wake models.

In Bingöl et al. (2010) the horizontal meandering has been examined with a nacelle-installed continuous wave (CW) LiDAR. The campaign confirms the passive tracer assumption, which is essential for the definition of the meandering in the DWM model. Furthermore, the wind speed deficit in the MFR has been investigated for some distances. Due to the fact that the CW LiDAR can not measure simultaneously in different downstream distances, the beam has been focused successively to different downstream distances. In Trujillo et al. (2011) the analysis has been extended to a two-dimensional scan. The measured wind
speed deficit in the MFR has been compared to the Ainslie wake model (Ainslie, 1988), which constitutes the basis of the deficit's definition in the DWM model.

Additionally, in Machefaux et al. (2013) a comparison of measured lateral wake meandering based on pulsed scanning LiDAR measurements has been presented. Special attention is paid to the advection velocity of the wake, which is estimated with measured and low-pass filtered wind directions at the metmast (based on the assumptions of the DWM model) and the
wake displacement at certain downstream distances. The analysis shows that the advection velocity calculated by the N.O. Jensen model is in relatively good agreement. Finally, the study compares the measured expansion of the wake in the fixed frame of reference (FFR) to CFD simulations and simple analytical engineering models. The wake expansion calculated by simple analytical engineering models is well in line with LiDAR measurements and CFD simulations, but also depicts potential for further improvements, which is why a new empirical model for single-wake expansion is proposed in Machefaux et al.
(2015). In Machefaux et al. (2016) a measurement campaign is presented, which involves three nacelle-mounted CW scanning LiDAR devices. The investigation includes a spectral analysis of the wake meandering, a comparison of the measurements to the assumptions in the DWM model as well as a comparison of the wind speed deficit profile in a merged wake situation to CFD simulations.

It should be noted that the references listed here are only the most essential, on which the present measurement campaign
builds up. Several campaigns including either LiDAR systems or meandering observations as well as wake model validations have been conducted in the past. The outlined analysis transfers some of the procedures of tracking the wake meandering to measurement results from an onshore wind farm with small turbine distances. Particular focus is put on the investigation of the wind speed deficit's shape in the MFR and the degradation of the wind speed deficit in downstream direction. The latter can be captured very well with the used nacelle-mounted pulsed scanning LiDAR systems due to the fact that it measures
simultaneously in different downstream distances. Thus, a detailed comparison of the predicted degradation of the wind speed deficit between the DWM model and the measurement results is possible. Furthermore, the collected LiDAR measurements are used to recalibrate the DWM model, which enables a more precise modeling of the wake degradation. As a consequence, the calculation of loads and energy yield of the wind farm can be improved.

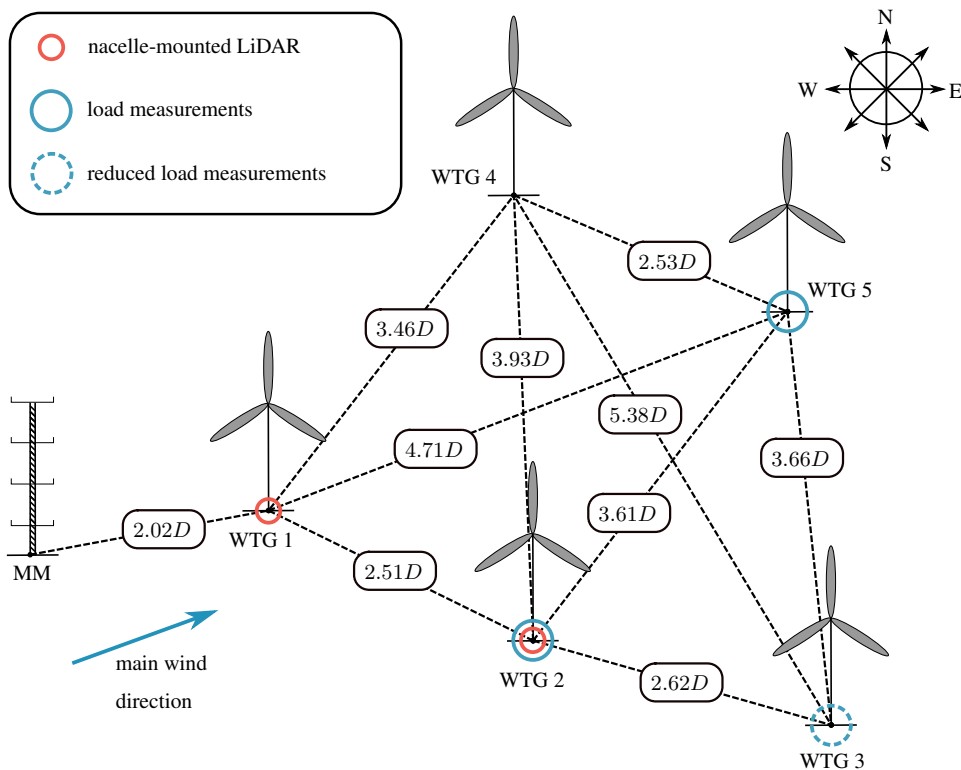

**Figure 1.** Wind farm layout with measurement equipment.

The remaining document is arranged as follows: In Section 2, the investigated wind farm and the installed measurement equipment are described in detail. Afterwards, in Section 3, an explanation of the data processing and filtering of the measurement results is given. Sections 4, 5, and 6 focus on the description of the theoretical backround and a hands-on implementation of the DWM model is introduced. Based on the outlined measurement results, a recalibration of the defined degradation of the wind speed deficit in the DWM model is proposed in Section 6. A summary of the measurement results can be found in Section 7 and a comparison to the original DWM model as well as the recalibrated version is presented in Section 8. Finally, all findings are concluded in Section 9.

## 2 Wind farm

The investigated onshore wind farm (Figure 1) located in the Southeast of Hamburg (Germany) consists of five closely spaced Nordex turbines (1x N117 3 MW and 4x N117 2.4 MW) and an IEC compliant 120 m metmast, which is situated two rotor diameters ($D = 117$ m) ahead of the wind farm in the main wind direction (west-southwest). It is equipped with 11 anemometers, two of which are ultrasonic devices, three wind vanes, two temperature sensors, two thermohygrometers, and two barometers. The sensors are distributed along the whole metmast, but at least one of each is mounted in the upper eight meters (see Figure

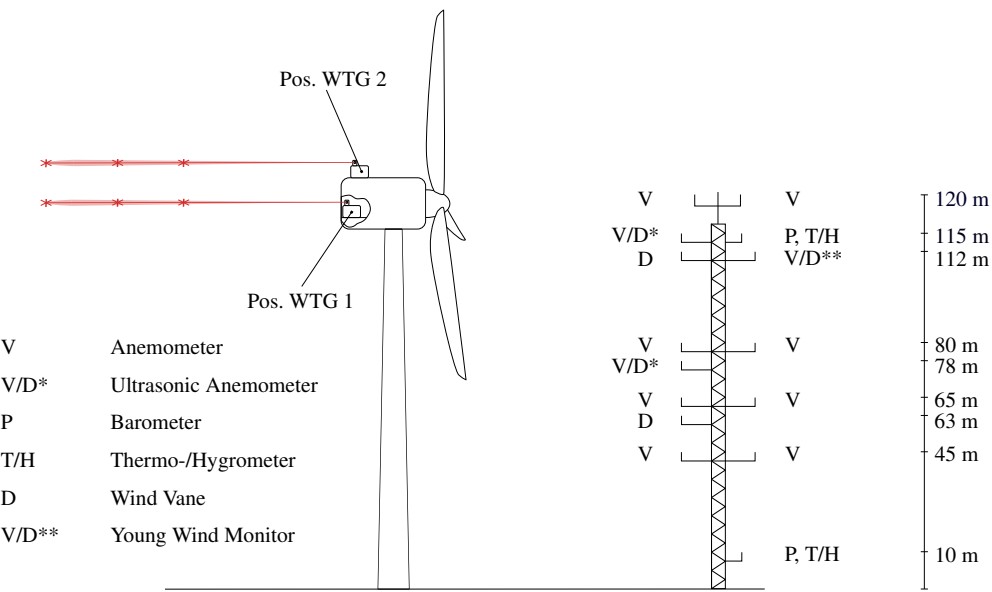

**Figure 2.** Metmast measurement equipment and LiDAR positions.

2). The thrust as well as the power coefficient curves for both wind turbines are illusatretd in Figure 3. There are no other tur-

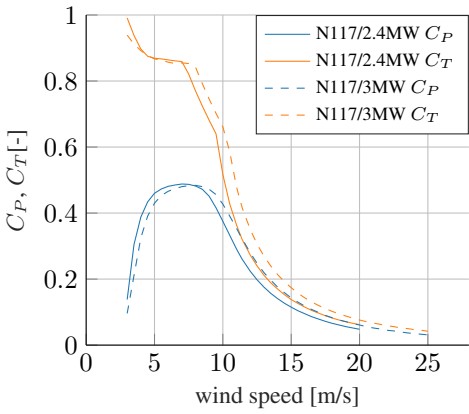

**Figure 3.** Power and thrust coefficients over wind speed for the N117/3MW and the N117/2.4MW turbines.

bines in the immediate vicinity and the terrain is mostly flat. Only in further distances (more than 1 km) the terrain is slightly hilly (approx. 40 m). Two turbine nacelles are equipped with a pulsed scanning LiDAR system (Galion G4000). The wind farm layout with all installed measurement devices is shown in Figure 1 (the displayed load measurements are not in the scope of this paper, but will be introduced in future publications). One LiDAR system is installed on top of the nacelle of WTG 2 (N117 2.4 MW), facing backwards. The second LiDAR system is installed inside the nacelle of WTG 1 (N117 3 MW) and measures through a hole in the rear wall. In this case, mounting the device on top of the nacelle is not possible, as the area is occupied

by a recuperator. The positions of both devices are displayed in Figure 2. Even though the setup reduces the field of vision, the measurement campaign described in this paper is not influenced by this restriction. On the plus side, the LiDAR system is not exposed to weather. Finally, nacelle-mounted differential GPS systems help tracking the nacelle's precise position as well as yaw movements with a centimeter range accuracy.

## 3  Data filtering and processing

The LiDAR data are filtered in accordance with the wind direction, so that LiDAR data without free inflow of the wake generating turbine as well as LiDAR measurements in the induction zone of another turbine are rejected. This leads to the remaining wind direction sectors listed in Table 1. The remaining sectors are relatively small, especially for the LiDAR on

**Table 1.** Considered wind direction sectors per wake generating turbine in the measurement campaign. Wind direction sectors without free inflow of the metmast and the turbine as well as measurements in the induction zone of another turbine are omitted.

|        | lower limit [°] | upper limit [°] |
|--------|-----------------|-----------------|
| WTG 1  | 160             | 190             |
|        | 320             | 350             |
| WTG 2  | 150             | 160             |
|        | 240             | 250             |

WTG 2, which reduces the amount of usable measurement data drastically. Additionally, the measured LiDAR data are sorted into turbulence intensity bins for the further validation and recalibration of the DWM model. The ambient conditions are determined by 10-minute time series statistics from the metmast, hence only measurement results with free inflow at the metmast are useable. Only situations with normal power production of the wake generating turbine are considered. The turbine operation mode is identified through the turbine's Supervisory Control and Data Acquisition (SCADA) system. The statistics of the 10-minute time series are applied to identify the operational mode. Furthermore, the data has been analyzed according to yaw misaligments, so that no data with turbine misalignments greater than $6°$ are considered in the analysis. The misalignment is determined by the GPS systems and the metmast wind direction. Moreover, the LiDAR data are filtered by the power intensity of the measurement results, which is closely related to the signal-to-noise ratio (SNR) of the measurements. Results with an intensity lower than 1.01 have been discarded. The pulse repetition rate of the LiDAR system is 15 kHz. The ray update rate is about 1 Hz (depending on the atmospheric conditions), so it averages over approximately 15000 pulses. The sample frequency is 100 MHz. Considering the speed of light, this delivers a point length of 1.5 m. The range gate length is 30 m, hence 20 points are used per range gate. The measurement time increases with the number of range gates, because the internal data processing time increases. Thus, to decrease the measurement time, the number of range gates has been limited, so that the farthest scan point is 750 m downstream. Additionally, the scanning time of each complete horizontal line scan is verified by the timestamp

of each scan to ensure that the meandering can really be captured. In summary, this leads to the following filtering procedure for the measured LiDAR data:

1. Filtering according to the wind direction determined by the metmast (free inflow at metmast and wind turbine and no induction zone from other turbines)

2. Filtering according to the normal power production determined by the turbine's SCADA system

3. Filtering according to yaw misaligment

4. Filtering according to the SNR of the LiDAR measurements

5. Filtering according to scan time

6. Grouping all data sets in turbulence intensity bins with a bin width of 2 %

LiDAR systems measure the line of sight velocity. The wind speed in downstream direction is then calculated from the LiDAR's LOS velocity and the geometric dependency of the position of the laser beam relative to the main flow direction as outlined in Machefaux et al. (2012). Thus, the horizontal wind speed is defined as

$$U(t) = U_{LOS} \cdot \frac{1}{\cos(\theta) \cdot \cos(\phi)} \; , \tag{1}$$

where $\theta$ is the azimuth angle and $\phi$ the elevation angle of the LiDAR scan head. This seems to be a suitable approach for small scan opening angles like in the measurement campaign presented here. The biggest opening angle in the scan pattern is 20°. Nevertheless, if there is yaw misalignment, this could have an impact on the overall results. To decrease the uncertainties based on yaw misalignments, the measurement data has accordingly been filtered. The yaw misalignment has the biggest impact at the largest scan opening angle, i.e., a misalignment of 6° at an opening angle of 20° leads to an overestimation of the wind speed of less than 5 %.

## 4 Wind speed deficit in HMFR calculation

The meandering time series and the wake's horizontal displacement are determined with the help of a Gaussian fit. Trujillo et al. (2011) assume that the probability of the wake position in vertical and horizontal direction is completely uncorrelated, so that the two-dimensional fitting function can be expressed as follows:

$$f_{2D} = \frac{A_{2D}}{2\pi\sigma_y\sigma_z} \exp\left[-\frac{1}{2}\left(\frac{(y_i - \mu_y)^2}{\sigma_y^2} + \frac{(z_i - \mu_z)^2}{\sigma_z^2}\right)\right] \; , \tag{2}$$

where $\sigma_y$ and $\sigma_z$ are the standard deviations of the horizontal and vertical displacements $\mu_y$ and $\mu_z$, respectively. In the analysis presented here, only results from a horizontal line scan are analyzed, so that no vertical meandering is eliminated from the wind speed deficit and the deficit's depth is less pronounced in comparison to the real MFR. To clarify that the vertical

meandering is not eliminated in the present investigation, but included in the wind speed deficit, the abbreviation HMFR (horizontal meandering frame of reference) is introduced and henceforth used instead of MFR. A comparison of the wind speed deficit simulated with the DWM model in the complete MFR and the HMFR is illustrated in Figure 4. The simulations were carried out for a small downstream distance of $2.5D$ and a high turbulence intensity of 16 %. There are only small discrepancies around the center of the wake, which validates the present assumption.

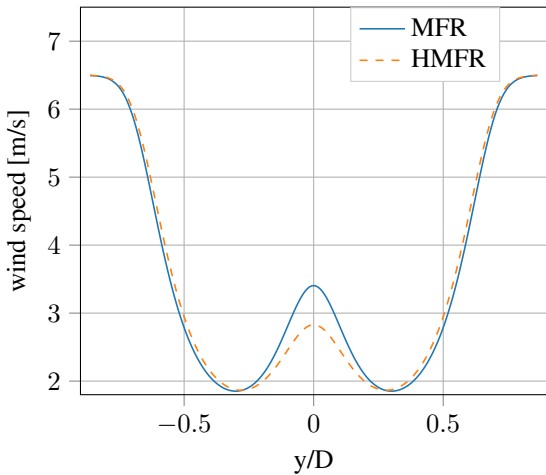

**Figure 4.** Wind speed deficit at a downstream distance of $2.5D$ and an ambient turbulence intensity of 16 %.

Since the vertical meandering is neglected, the measurement results are fitted to a one-dimensional Gaussian curve defined as follows:

$$f_{1D} = \frac{A_{1D}}{\sqrt{2\pi}\sigma_y} \exp\left(-\frac{1}{2}\frac{(y_i - \mu_y)^2}{\sigma_y^2}\right) , \tag{3}$$

where $A_{1D}$ represents a scaling parameter. The measured wind speeds are fitted to the Gauss shape via a least-squares method. Thereby, only fitted horizontal displacements $\mu_y$ that are between -200 m and 200 m are used for further validations of the mean wind speed in the HMFR. A horizontal displacement of more than 200 m cannot be represented by the Gauss fit due to a lack of measurement points. However, such an event is highly improbable (e.g., the DWM model predicts the wind speed deficit's probability at the horizontal position of 200 m to be $2 \cdot 10^{-22}$ for an ambient wind speed of 6.5 m/s and an ambient turbulence intensity of 8 %).

The entire method of calculating the wind speed deficit in the HMFR is illustrated in Figure 5 and can be described as follows: The LiDAR system takes measurements from the nacelle of the turbine in downstream direction, which deliver the wind speed deficit in the nacelle frame of reference or even in the FFR (see left side of Figure 5) if the turbine is not moving (this is ensured by the GPS systems). A Gauss curve is then fitted into the scanned points as explained previously. It provides the horizontal displacement of the wake, so that each scan point can be transferred into the HMFR with the calculated displacement (see middle diagrams in Figure 5). The last step illustrated in the diagrams is the interpolation to a regular grid. These three

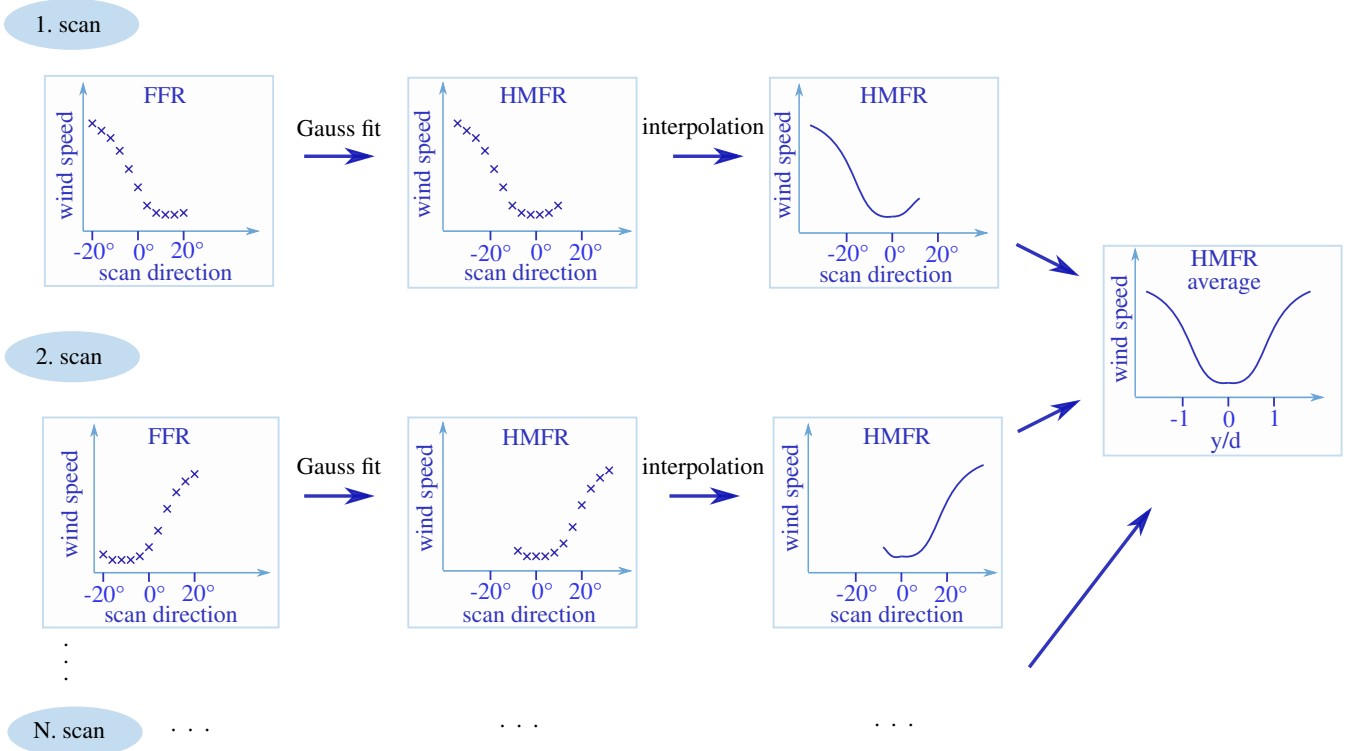

**Figure 5.** Method for the determination of the mean wind speed deficit in the HMFR.

steps are repeated for a certain number of scans N (e.g., approx. 37 for a 10-min time series). Finally, the mean value of all
single measurement results in the HMFR is calculated. It should be noted that it is mandatory to interpolate to a regular grid.
Otherwise it would not be possible to take the mean of all scans since the horizontal displacement differs at each instant of time
and, thereupon, the measurement points are transmitted to a different location in the HMFR. After averaging, the plausibility
of the results is inspected. If the calculated minimum mean wind speed in the HMFR is higher than the minimum mean wind
speed in the FFR, it is assumed that the Gauss fit failed and the results are no longer considered. In theory, the wind speed
deficit in the HMFR should be more pronounced than the measured one in the FFR, wherefore this fundamental plausibilty
check is added.

## 5   LiDAR simulation

One of the most challenging parts of this specific measurement campaign is the low ray update rate of the LiDAR system, which
is considerably smaller than in the previously introduced measurement campaigns (Bingöl et al., 2010; Trujillo et al., 2011).
To ensure that the meandering as well as the wind speed deficit in the HMFR can be captured with the devices used, LiDAR
and wind field simulations have been conducted in advance. The simulations incorporate LiDAR specifications (e.g., beam

update rate and scan head angular velocity) and wind farm site conditions (ambient turbulence intensity and wind shear). The simulations assume perfect LiDAR measurements, where no probe volume averaging is considered and the LiDAR measures the horizontal wind speed directly. The wind field is simulated at halfway of the range gate. The simulated LiDAR "takes measurements" in a simulated wind field that is generated by the DWM model and includes wake effects as well as ambient turbulences. A detailed description of the model is given in Section 6. The in-house code is written in Python. From these "measured" wind speeds the meandering is determined via Gaussian fits as previously explained and implemented in the real measurement campaign. Simulations are performed for different scan patterns, ambient conditions, and downstream distances to test the scan pattern, which for this one-dimensional scan consists of only 11 scan points scanned in a horizontal line from $-20°$ to $20°$ in $4°$ steps. The "measurement" results of the simulated meandering time series are shown in Figure 6(a), whereas the corresponding wind speed deficit in the HMFR is presented in Figure 6(b). The results are compared to the original

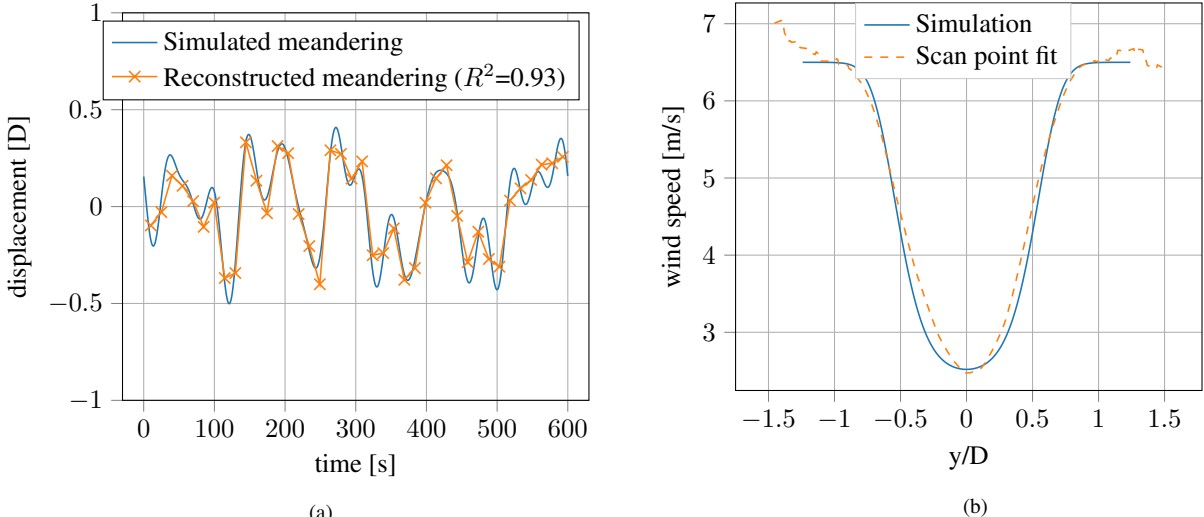

(a)

(b)

**Figure 6.** Simulated and simulated "measured" meandering time series **(a)** and wind speed deficit in the HMFR **(b)** at an ambient wind speed of 6.5 m/s.

meandering time series and the simulated wind speed deficit. The "measured" wind speed deficit in the simulated environment reproduces the simulated wind speed and its underlying meandering time series very well (the coefficient of determination $R^2$ is approximately 0.93). Although only 11 scan points are used for these plots, the curve of the wind speed deficit is very smooth. The reason for this behavior is the previously mentioned interpolation process. The distribution generated by the meandering process provides many scan points around the center of the wind speed deficit and only a few at the tails. Therefore, the influence of turbulence at the tails is much higher, leading to a somewhat coarse distribution at the boundaries of the deficit. It should also be noted that since this is a one-dimensional scan, the simulated LiDAR "measures" the wind speed deficit only horizontally neglecting the wake's less dominant vertical movement. Whenever the wind speed deficit in the

180 HMFR is mentioned in subsequent validations, it implies the neglection of eliminating the vertical meandering from the wind speed deficit, which has only a marginal impact on the shape of the wind speed deficit in the real MFR (see Figure 4).

The LiDAR simulations indicate that the Gauss fit works more reliably under optimal operating conditions, i.e., at optimal tip speed ratio, when the wind speed deficit is most pronounced and the power coefficient $C_p$ has its maximum (see Figure 3). For the turbines examined, this applies to a range of 5 m/s up to 8 m/s, so that only measurement results with ambient wind speeds in this interval are analyzed.

## 6 Dynamic wake meandering model

The measured wind speed deficit in the HMFR is consecutively compared to the DWM model, which is based on the assumption that the wake behaves as a passive tracer in the turbulent wind field. Consequently, the movement of the passive structure, i.e., the wake deficit, is driven by large turbulence scales (Larsen et al., 2007, 2008b). The main components of the model are summarized in Figure 7(a). The model was built in-house and independent from any commercial software in Python.

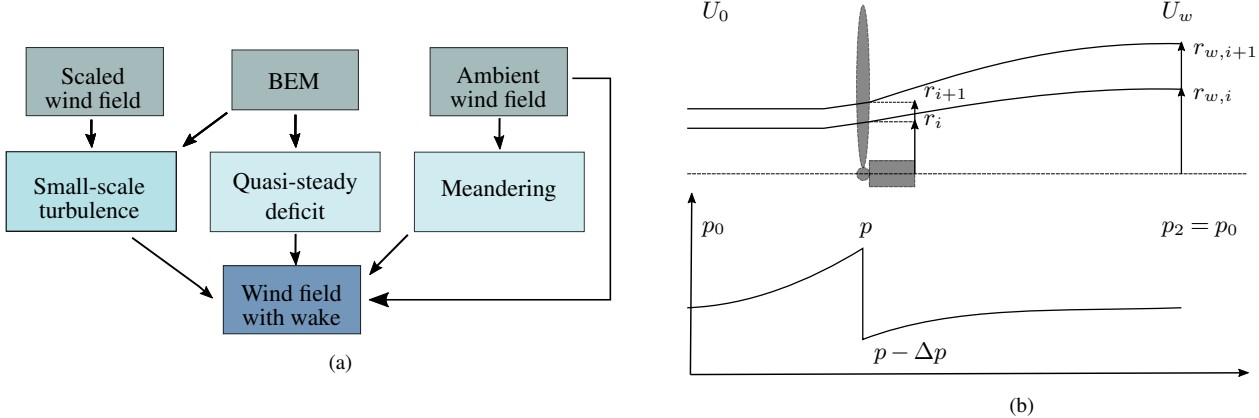

**Figure 7.** Components of the DWM model **(a)** (Reinwardt et al., 2018) and schematic illustration of the wake expansion in the DWM model **(b)** according to Madsen et al. (2010).

### 6.1 Quasi-steady wake deficit

One key point of the model is the quasi-steady wake deficit or rather the wind speed deficit in the MFR. In this study, two calculation methods for the quasi-steady wake deficit are compared with the LiDAR measurement results. A similar comparison of these models to metmast measurements in the FFR was published in Reinwardt et al. (2018). The quasi-steady wake deficit

is defined in the MFR and consists of a formulation of the initial deficit emitted by the wake generating turbine and the expansion of the deficit downstream (Larsen et al., 2008a). The latter is calculated with the thin shear layer approximation of the Navier-Stokes equations in their axisymmetric form. This method is strongly related to the work of Ainslie (1988) and

outlined in Larsen et al. (2007). The thin shear layer equations expressed by the wind speed in axial and radial direction $U$ and $V_r$, respectively, are defined by

$$U\frac{\partial U}{\partial x} + V_r\frac{\partial U}{\partial r} = \frac{1}{r}\frac{\partial}{\partial r}\left(\nu_T r\frac{\partial U}{\partial r}\right) \tag{4}$$

and

$$\frac{1}{r}\frac{\partial}{\partial r}(rV_r) + \frac{\partial U}{\partial x} = 0 . \tag{5}$$

The first part of the quasi-steady wake deficit, the initial deficit, serves as a boundary condition when solving the equations. In both methods used to determine the quasi-steady wake deficit, the initial deficit is based on the axial induction factor derived from the blade element momentum (BEM) theory. Pressure terms in the thin shear layer equations are neglected. The error that inherently comes with this assumption is accommodated by using the wind speed deficit two rotor diameters downstream (beginning of the far-wake area) as a boundary condition for the solution of the thin shear layer equations. The equations are solved directly from the rotor plane by a finite-differences method with a discretization in axial and radial direction of $0.2D$ and $0.0125D$ combined with an eddy viscosity ($\nu_T$) closure approach. The two methods that are compared with the LiDAR measurements only differ in the definition of the initial deficit and the eddy viscosity formulation.

### 6.1.1 DWM-Egmond

For the first method the following formulae are given to calculate the initial deficit. Hence, the boundary condition for solving the thin shear layer equations are (Madsen et al., 2010):

$$U_w\left(\frac{r_{w,i+1} + r_{w,i}}{2}\right) = U_0(1 - 2a_i) \tag{6}$$

and

$$r_{w,i+1} = \sqrt{\frac{1-a_i}{1-2a_i}\left(r_{i+1}^2 - r_i^2\right) + r_{w,i}^2}\, f_w \tag{7}$$

with

$$f_w = 1 - 0.45\bar{a}^2 , \tag{8}$$

where $\bar{a}$ represents the mean induction factor along all radial positions $i$, $r_i$ the rotor radius and $r_{w,i}$ the wake radius. The boundary condition of the radial velocity component is $V_r = 0$. The initial wake expansion and the corresponding radial positions as well as the pressure recovery in downstream direction are illustrated in Figure 7(b). The eddy viscosity $\nu_T$ used in Equation (4), is calculated in this first approach as follows (Larsen et al., 2013):

$$\frac{\nu_T}{U_0 R} = k_1 F_1(\tilde{x})F_{amb}(\tilde{x})I_0 + k_2 F_2(\tilde{x})\frac{R_w(\tilde{x})}{R}\left(1 - \frac{U_{min}(\tilde{x})}{U_0}\right) \tag{9}$$

with $k_1 = 0.1$ and $k_2 = 0.008$. The eddy viscosity is normalized by the ambient wind speed $U_0$ and the rotor radius $R$. The outlined definition consists of two terms. The first is related to the ambient turbulence intensity $I_0$, whereas the second depends

on the shape of the wind speed deficit itself. The single terms are weighted with the factors $k_1$ and $k_2$. The filter functions $F_1$ and $F_2$ in Equation (9) depending on $\tilde{x}$ (downstream distance normalized by the rotor radius) are defined by IEC 61400-1 Ed.4 as follows:

$$F_1(\tilde{x}) = \begin{cases} \left(\frac{\tilde{x}}{8}\right)^{3/2} - \frac{\sin\left(\frac{2\pi\tilde{x}^{3/2}}{8^{3/2}}\right)}{2\pi} & \text{for } 0 \leq \tilde{x} < 8 \\ 1 & \text{for } \tilde{x} \geq 8 \end{cases} \tag{10}$$

and

$$F_2(\tilde{x}) = \begin{cases} 0.0625 & \text{for } 0 \leq \tilde{x} < 4 \\ 0.025\tilde{x} - 0.0375 & \text{for } 4 \leq \tilde{x} < 12 \\ 0.00105(\tilde{x} - 12)^3 + 0.025\tilde{x} - 0.0375 & \text{for } 12 \leq \tilde{x} < 20 \\ 1 & \text{for } \tilde{x} \geq 20 \,. \end{cases} \tag{11}$$

The filter function $F_2$ covers the lack of equilibrium between the velocity field and the rising turbulence in the beginning of the wake. $F_1$ is introduced to include the fact that the depth of the wind speed deficit increases in the near-wake area up to $(2...3)D$ downstream of the turbine until it attenuates again in downstream direction (Madsen et al., 2010). The filter function as well

as Equation (8) are calibrated against actuator disc simulations at a downstream distance of $2D$, the beginning of the far-wake area, where the wake is fully expanded (Madsen et al., 2010). A more detailed explanation of the nonlinear coupling function $F_{amb}$ is given in Section 6.3. This calculation method (Equations (6) to (11)) is subsequently named "DWM-Egmond" after the site, which is used for the calibration of the eddy viscosity in Larsen et al. (2013).

### 6.1.2   DWM-Keck

The second investigated method defines the initial deficit by the following equations (Keck, 2013):

$$U_w(r_{w,i}) = U_0 \left(1 - (1 + f_u) a_i\right) \tag{12}$$

and

$$r_{w,i} = r_i \sqrt{\frac{1 - \bar{a}}{1 - (1 + f_R)\bar{a}}} \tag{13}$$

with $f_u = 1.1$ and $f_R = 0.98$. The boundary condition of the radial velocity component is again $V_r = 0$. In Keck (2013) the

final and recommended version of the model developed for the eddy viscosity is defined as follows:

$$\nu_T = k_1 F_1(\tilde{x}) \, u^*_{ABL;\lambda<2D} \, l^*_{ABL;\lambda<2D} + k_2 F_2(\tilde{x}) \max\left(l^{*2}\left|\frac{\partial U(\tilde{x})}{\partial r}\right|, l^* \left(1 - U_{min}(\tilde{x})\right)\right) \tag{14}$$

with $k_1 = 0.578$ and $k_2 = 0.0178$ and the filter functions:

$$F_1 = \begin{cases} \frac{\tilde{x}}{4} & \text{for } \tilde{x} < 4 \\ 1 & \text{for } \tilde{x} \geq 4 \end{cases} \tag{15}$$

and

$$F_2 = \begin{cases} 0.035 & \text{for } \tilde{x} < 4 \\ 1 - 0.965 e^{-0.35(\tilde{x}/2 - 2)} & \text{for } \tilde{x} \geq 4 \, . \end{cases} \tag{16}$$

In contrast to the previously mentioned model (DWM-Egmond) atmospheric stability is considered in this final model description. Equation (14) involves the velocity $u^*_{ABL;\lambda<2D}$ and length scale $l^*_{ABL;\lambda<2D}$ fractions of the ambient turbulence, which is related to the wake deficit evolution (eddies smaller than $2D$). The velocity scale $u^*_{ABL;\lambda<2D}$ is besides the ambient turbulence intensity $I_0$ related to the ratio of the Reynolds stresses (normal stress in flow direction and the shear stress), which in turn are functions of the atmospheric stability. A detailed description of a method to introduce atmospheric stability in the DWM model can be found in Keck et al. (2014) and Keck (2013). In contrast to the final and recommended model in Keck (2013), atmospheric stability is not considered in this study, so that a previous model in Keck (2013) without consideration of atmospheric stability is used and the numerical constants $k_1$ and $k_2$ in Equation (17) are changed with respect to the first least-squares recalibration in Keck (2013). Furthermore, according to Keck (2013) it can be assumed that the mixing length $l^*$ is equal to half of the wake width. This results in the following formulation of the eddy viscosity:

$$\frac{\nu_T}{U_0 R} = k_1 F_1(\tilde{x}) I_0 + k_2 F_2(\tilde{x}) \max\left( \frac{R_w(\tilde{x})^2}{R U_0} \left| \frac{\partial U(\tilde{x})}{\partial r} \right|, \frac{R_w(\tilde{x})}{R} \left( 1 - \frac{U_{min}(\tilde{x})}{U_0} \right) \right) \tag{17}$$

with $k_1 = 0.0914$ and $k_2 = 0.0216$.

## 6.2 Meandering of the wake

The meandering of the wind speed deficit is calculated from the large turbulence scales of the ambient turbulent wind field. Thus, the vertical and horizontal movements are calculated from an ideal low-pass filtered ambient wind field. The cut-off frequency of the low-pass filter is specified by the ambient wind speed and the rotor radius as (Larsen et al., 2013):

$$f_c = \frac{U_0}{4R} \, . \tag{18}$$

The horizontal $y(t)$ and vertical $z(t)$ positions of the wind speed deficit are calculated based on the low-pass filtered velocities in horizontal and vertical directions according to the relations (Larsen et al., 2007):

$$\frac{\mathrm{d}y(t)}{\mathrm{d}t} = v(t) \tag{19}$$

and

$$\frac{\mathrm{d}z(t)}{\mathrm{d}t} = w(t), \tag{20}$$

where $v(t)$ and $w(t)$ are the fluctuating wind speeds at hub height. The ambient wind field, which is later on low-pass filtered, is generated in this work by a Kaimal spectrum and a coherence function (e.g., Veers, 1988). The temporal resolution of the generated wind field is 0.07 s.

### 6.3  Recalibration of the DWM model

The wind speed deficit measured by the LiDAR systems is used to recalibrate the wake degradation downstream or to be more precise the eddy viscosity description. In Larsen et al. (2013) a recalibration was already achieved by introducing a nonlinear coupling function $F_{amb}$ into the ambient turbulence intensity term of the eddy viscosity definition (see Equation (9)). Furthermore, a comparison between the measured and simulated power based on the DWM model was carried out. It shows that the wind speed deficit degradation is too low for lower turbulence intensities and moderate to high turbine distances in the model version from Madsen et al. (2010). For this reason, the downstream distance dependent function $F_{amb}$ was introduced into the eddy viscosity description in Larsen et al. (2013).

A similar behavior but even more pronounced can be seen in the results in Section 7. Following the approach of Larsen et al. (2013), a function based on a least-squares calibration with the acquired LiDAR measurements is developed. This function is incorporated into the normalized eddy viscosity description in Eq. (17), whereby it changes to:

$$\frac{\nu_T}{U_0 R} = k_1 F_{amb}(\tilde{x}) F_1(\tilde{x}) I_0 + k_2 F_2(\tilde{x}) \max\left( \frac{R_w(\tilde{x})^2}{RU_0} \left| \frac{\partial U(\tilde{x})}{\partial r} \right|, \frac{R_w(\tilde{x})}{R} \left( 1 - \frac{U_{min}(\tilde{x})}{U_0} \right) \right) \tag{21}$$

with the constants $k_1 = 0.0924$ and $k_2 = 0.0216$ and the coupling function

$$F_{amb}(\tilde{x}) = a\tilde{x}^{-b} \tag{22}$$

with $a = 0.285$ and $b = 0.742$. The parameters a and b are the results of the least-squares calibration. It should be noted that the constant $k_1$ was also slightly adjusted by the recalibration, in which the normalized eddy viscosity definition of Keck (2013) has been used. This derives from the fact that this model is already in good agreement with the measurement results in most turbulence intensity bins as demonstrated in Section 8 and also in Reinwardt et al. (2018).

## 7  Measurement results

The measurement campaign lasted from January to July 2019. Both LiDAR systems, introduced in Section 2, were used to collect the data. Results of the meandering time series over 10 minutes are exemplarily shown in Figure 8(a). The maximum displacement of the wake is about $0.5D$, which is equivalent to 58.5 m. The results are derived from a 10-min time series with an ambient wind speed of 6.44 m/s and an ambient turbulence intensity of 11.7 %. Some of the metmast detected ambient conditions (wind speed $U_0$, turbulence intensity $I_0$, wind shear $\alpha$ and wind direction $\theta$) are given in the title of the figure. The corresponding mean wind speed deficit is illustrated in Figure 8(b). The wind speed decreases to less than 3 m/s in full wake situations. As explained in Section 5, the tails of the curve are relatively coarse since less scan points were gathered. It can also be seen that the ambient wind speed is not even reached at the edges of the curve. The opening angle of the scan appears too small to capture the whole wake at this distance. Towards the left part of the wind speed deficit (at negative y distances) a bigger part of the wake is captured. This arises from the fact that the horizontal displacement is more often positive than negative and, therefore, more measurement results are collected towards the left part of the wind speed deficit curve.

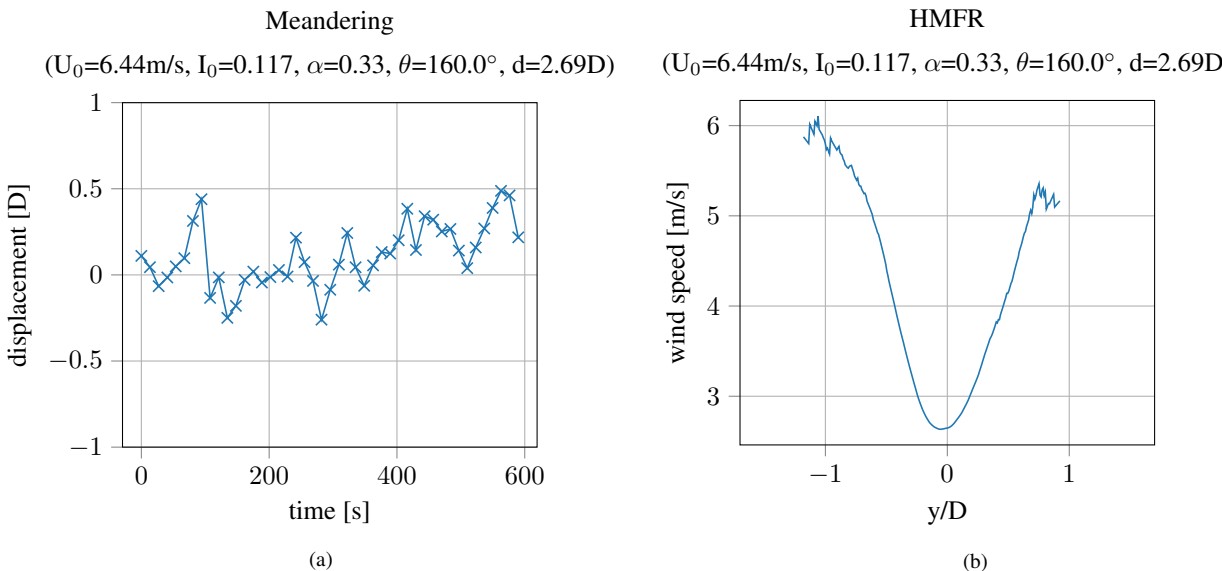

**Figure 8.** Meandering time series **(a)** and wind speed deficit in the HMFR **(b)** at $2.69D$ downstream of the turbine.

The used LiDAR system is capable of measuring several range gates simultaneously in $30\,\mathrm{m}$ intervals. The results of all detected range gates for the data set presented in Figure 8 are shown in Figure 9(a). The closest distance is $1.92D$ downstream

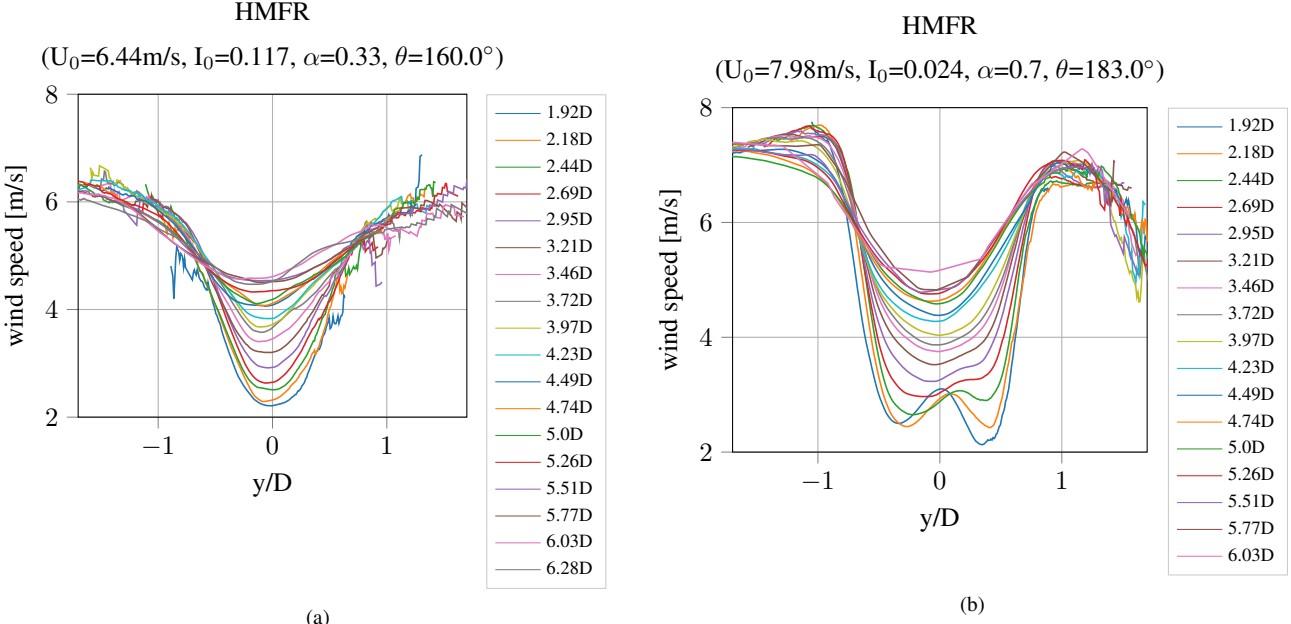

**Figure 9.** Wind speed deficit in the HMFR for an ambient turbulence intensity of 11.7 % **(a)** and a turbulence intensity of 2.4 % **(b)**.

and the farthest is $6.28D$. The degradation of the wind speed deficit in downstream direction is clearly identifiable. As for the single distance case (Figure 8), for most range gates a bigger database is captured at the left part of the wind speed deficit, resulting in smoother curves. The presumption of a too small opening angle of the scan, as stated before, proves true. With increasing downstream distances the captured wind speed deficits get closer to integrity. A broader scan angle would result in more detailed wind speed deficits for close downstream distances at the expense of far distances, where the scan points might not capture enough points inside the deficit and thereby prevent a successful Gaussian fit. Furthermore, additional scan points at the edges can lead to a better representation of the deficit but would also increase the scan time. According to Equation (18), the meandering is correlated to frequencies lower than approximately 0.028 Hz considering a wind speed of 6.5 m/s and a rotor diameter of 117 m. This means that, considering the Nyquist–Shannon sampling theorem, the scan time must be longer than half of the reciprocal of 0.028 Hz, which results in a neccessary scan time of less than 18 s. The scan time for the current usage of 11 scan points is already at about 16 s (depending on the visibility conditions), which is close to the limit of 18 s, so with an increased number of scan points it is no longer ensured that the meandering can be captured.

Figure 9(b) illustrates the wind speed deficit in the HMFR measured under different ambient conditions. The corresponding meandering time series and wind speed deficit for this measured time series at $2.69D$ downstream is given in Figure A1 in the appendix. The wind shear is fairly high ($\alpha = 0.7$) and the turbulence intensity is very low ($I_0 = 2.4$ %). Due to the low turbulence intensity it is still possible to see the w-shape of the wind speed deficit at closer distances. The typical w-shape is caused by the low axial induction in the area of the nacelle. Further downstream, the wake becomes more Gaussian shaped. At a horizontal distance of about $1.5D$ from the wake center, the wind speed decreases. The reason is the wake of other turbines in the wind farm. The mean wind direction in this time series is 183° and the measurements are taken from WTG 1, so it could be either the influence of the wakes of WTG 2 or WTG 4. The associated results of the mean wind speed deficit in the FFR are illustrated in Figure 10. The curves in the FFR are less smooth than the wind speed deficit in the HMFR, simply because only 11 points are scanned and no interpolation is necessary when calculating the mean wind speed over the whole time series. Comparing Figures 9 and 10, it becomes apparent that the wind speed deficit in the FFR is less pronounced. Furthermore, for the lower turbulence intensity the w-shape of the wind speed is not visible, since it vanished due to the meandering.

Similar results as exemplarily shown in Figures 9 and 10 have been collected for a multitude of different ambient conditions. The number of measured time series per turbulence intensity and wake generating turbine, on which the LiDAR system is installed, is listed in Table 2. The turbulence intensity is binned in 2° steps. Column 1 of Table 2 specifies the mean values for each bin. Most of the measurement results are collected at low to moderate turbulence intensities ($I_0 = (4 - 10)$ %). Only a few results could be extracted at higher turbulence intensities. The results include time series with an ambient wind speed of 5 m/s to 8 m/s. In this range, both turbines operate under optimal and most efficient conditions resulting in maximum energy output from the wind. The thrust coefficient is constant in this region (see Figure 3). Therefore, the axial induction and the wind speed deficit normalized by the turbine's inflow wind speed are also expected to be constant for similar ambient conditions over this wind speed range. For the single turbulence intensity bins and both turbine types, simulations with different DWM models are carried out applying the same axial induction over the whole wind speed range. A scatterplot of the shear exponent and the

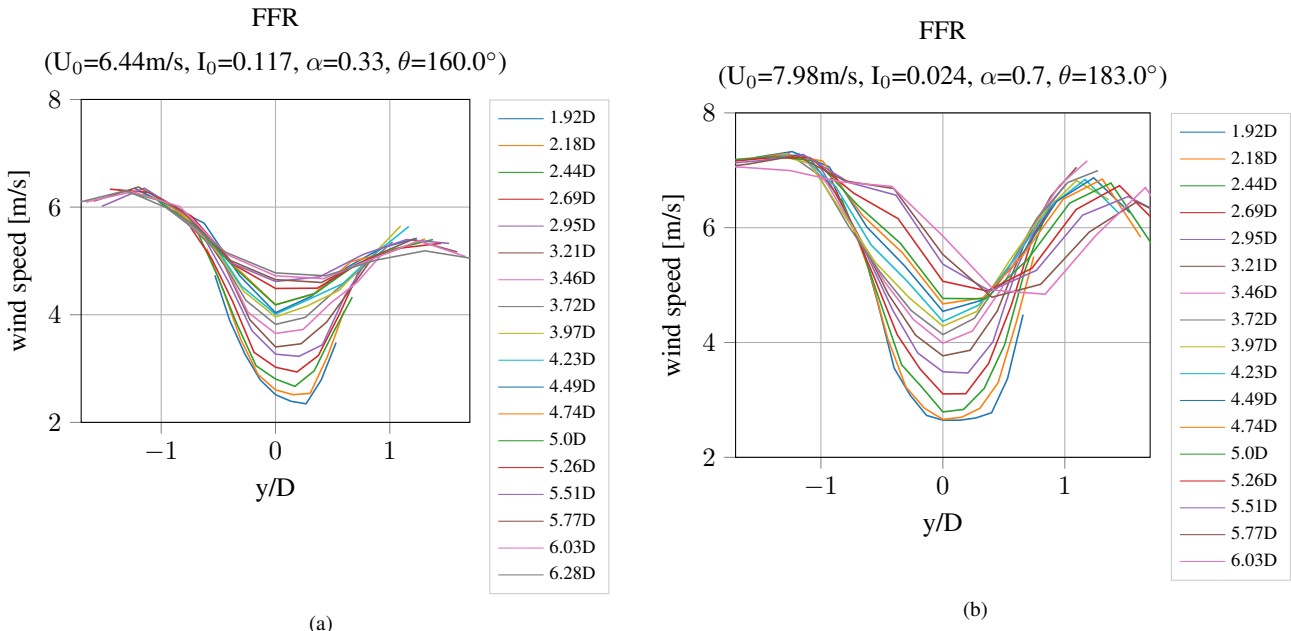

**Figure 10.** Wind speed deficit in the FFR for a turbulence intensity of 11.7 % **(a)** and a turbulence intensity of 2.4 % **(b)**.

**Table 2.** Number of measured and considered data sets per turbulence intensity for the LiDAR systems on WTG 1 and WTG 2.

| $I_0$ [%] | WTG 1 | WTG 2 |
|---|---|---|
| 4 | 23 | 28 |
| 6 | 8 | 11 |
| 8 | 23 | 14 |
| 10 | 11 | 9 |
| 12 | 13 | 4 |
| 14 | 0 | 0 |
| 16 | 1 | 1 |
| 18 | 1 | 2 |
| 20 | 1 | 3 |
| 22 | 0 | 2 |

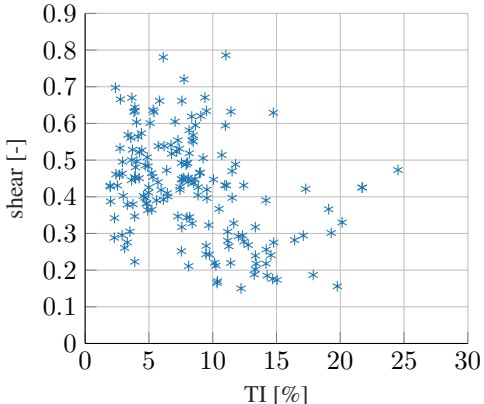

**Figure 11.** Shear exponent over the ambient turbulence intensity for all considered data sets.

ambient turbulence intensity determined by the metmast is given in Figure 11. It includes all used datasets. At lower turbulence intensities, the shear spreads quite a lot, whereas towards higher turbulence intensities the shear decreases as expected.

Figure 12 summarizes all measured wind speed deficits in the HMFR. It demonstrates the mean value and the standard

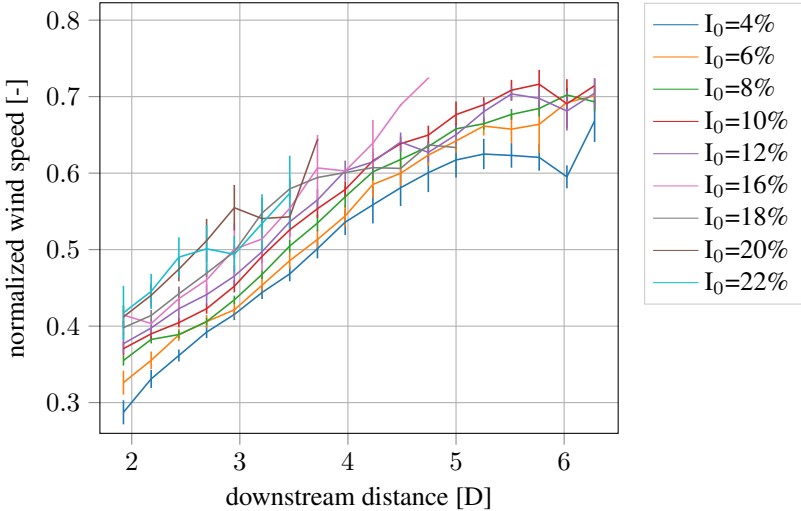

**Figure 12.** Measured mean value (line) and standard deviation (bar) of the mean value of the minimal wind speed in the HMFR for different turbulence intensity bins with a bin width of 2 %.

deviation of the mean for all captured turbulence bins plotted against the downstream distance. Each value is related to the minimum value of the wind speed deficit, which itself is normalized by the inflow wind speed. It should be noted that in some distances only one value satisfies the filtering and plausibility checks, whereby the error bar is omitted. Additionally, it is pointed out that the plotted values always refer to the minimum value of a wind speed curve and not necessarily to the velocity in the wake center. Therefore, no increase of the wind speed at low downstream distances on account of the w-shape is visible. The

wind speed deficit at the wake center plotted against the downstream distance is depicted in the next section in Figure 15(b) and will be discussed further at this point. Figure 12 illustrates very well that the lowest degradation of the wind speed deficit occurs at the lowest turbulence intensity. Up to a turbulence intensity of 10 %, the degradation of the wind speed deficit continuously rises, leading to increasing minimum wind speeds at nearly all downstream distances. Above 10 % turbulence intensity, the case is less clear. Especially at larger downstream distances, the measured normalized minimum wind speed happens to fall

below the corresponding lower turbulence intensity bin. An explanation is the reduced number of measurement results in these bins and the higher uncertainty that comes along with it (expressed as error bars). Furthermore, discrepancies in the determined ambient turbulence intensity at the metmast location and the actual turbulence intensity at the wake position could lead to a misinterpretation of the LiDAR measurements. The farthest distance between the metmast and the location measured by the LiDAR system that occurs in the analyzed sectors is about 1200 m. With an ambient wind speed of 6.5 m/s, this leads to a wake

advection time of 185 s, thus even at worst conditions, the measured ambient conditions at the metmast should be valid for

the measured wakes from the LiDAR system most of the time. Furthermore, there is no complex terrain at the site, so it can be assumed that the conditions do not change with the wind direction. In addition, the agreement between measurements and simulations is already good in the higher turbulence intensity bins. Thus, the recalibration affects only the lower turbulence intensity bins with larger amounts of data, while the influence of the calibration on higher turbulence intensities is negligible (see Figure 13). Therefore, even though there are some discrepancies, the faster recovery of the wind speed deficit due to the higher ambient turbulence intensity can be verified and the measurements are reliable for the outlined investigation. Thus, it is valid to use these measurement results for comparisons with DWM model simulations and the recalibration of the DWM model in the next section.

## 8 Comparison between measurements and DWM model simulation

Figure 13 compares the measured normalized minimum wind speed in the wake to DWM model simulations. The left part of Figure 13 shows results for a relatively low turbulence intensity of 6 %, whereas the right part contains results for a higher turbulence intensity of 16 %. Further results for the remaining turbulence intensity bins are shown in Figures B1 and B2 in the appendix. The simulations were carried out for a specific downstream distance, which corresponds to the center of the range gate of the LiDAR system. It should be noted that the wind speeds measured by the LiDAR system can be interpreted as a mean value over the whole range gate. However, the wind speed gradient in axial direction is low and almost linear in the observed downstream distances, so even in the DWM model, the discretization in downstream direction is 23.4 m (equivalent to $0.2D$), which is in the same order of the range gate of 30 m. Therefore, a valid comparison between simulations and measurements is carried out. The wind speed deficit simulations in the HMFR obtained by the DWM model also include the vertical meandering to ensure a correct comparison between measurements and simulations. Three different simulation results with varying definitions of the initial deficit and eddy viscosity description are illustrated. The method called "DWM-Egmond" is based on the definitions of Madsen et al. (2010) and Larsen et al. (2013) and the "DWM-Keck" method is adopted from Keck (2013), see Section 6. Figure 13 shows that the DWM-Egmond method overestimates the wind speed deficit for all downstream distances and for both turbulence intensities. The simulated minimum wind speed with the DWM-Keck method is in better agreement with the measurement results. This confirms the results in Reinwardt et al. (2018). Especially at higher turbulence intensities (Figure 13), the results of the DWM-Keck model agree very well with the measurements. For lower turbulence intensities and higher distances (greater than $3D$) there is a relatively large discrepancy between measurements and simulations. A similar observation was made in Larsen et al. (2013) with the model version in Madsen et al. (2010). Aiming at the adjustment of the simulated degradation of the wind speed deficit in Larsen et al. (2013) for cases like the one presented here, the DWM model has been recalibrated and is henceforth called "DWM-Keck-c" (see Figure 13).

The recalibration of the DWM model and accordingly the normalized eddy-viscosity definition in the DWM model are based on a least-squares fit of the minimum of the simulated normalized wind speed to the minimum of the measured normalized wind speed for several downstream distances. The definition of the eddy viscosity along with the recalibrated parameters are explained in detail in Section 6.3. For the recalibration the measurement results are divided into 2 % turbulence intensity bins.

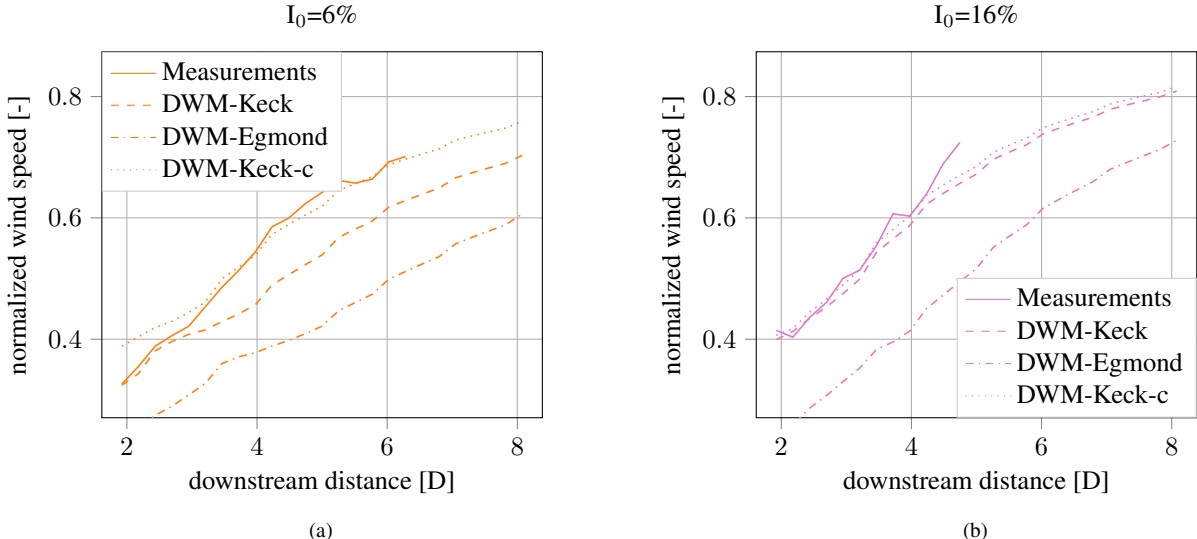

**Figure 13.** Comparison of measurements and simulations of the minimum wind speed deficit in the HMFR for different turbulence intensities. The recalibrated model is denoted DWM-Keck-c.

All measurement results from Figure 12 containing data sets from two different turbines, are used for the recalibration. The first turbine is an N117 turbine with 3 MW and the second one is an N117 with 2.4 MW. DWM model simulations were carried out for both turbine types, since the axial induction of both turbines is slightly different under partial load conditions. To calculate a mean value of the simulated minimum wind speed and thus allow a comparison with the results in Figure 12, simulations with both turbine types are carried out for each turbulence intensity bin and weighted in accordance with the number of measurement results per turbine listed in Table 2. Thus, for example at the ambient turbulence intensity bin of 4 %, the mean value of the simulated minimum wind speed consists of the sum of the simulated minimum wind speeds weighted by 0.451 and 0.549, the weighting factors for WTG1 and WTG2, respectively. Nonetheless, this weighting has only a marginal influence on the overall results, because the axial induction in the considered wind speed range (5 m/s to 8 m/s) is very similar for these two turbine types (see also thrust and power curves in Figure 3).

The results of the recalibrated DWM model, denoted Keck-c in Figure 13, coincide very well with the measurements. In particular, the results for lower turbulence intensities could clearly be improved. For higher turbulence intensities, the influence of the recalibration is less significant and the already good agreement between simulation and measurement results remains unchanged. The same applies to the results in the appendix in Figures B1 and B2. Only at the lowest downstream distances and turbulence intensities up to 12 %, the recalibrated model delivers higher deviations than the original model. For downstream distances larger than $3D$, the recalibrated model leads to more than 10 % lower devitaions from the measurements than the original model. For turbulence intensities higher than 16 %, the deviation between the recalibrated and original model is smaller than the uncertainties in the measurements, hence no further conclusions about improvements can be made. The uncertainties

in accordance to misalignments could be up to 5 % (see also the data filtering in Section 3). Furthermore, the LOS accuracy of the LiDAR system itself is about 1.5 % at a wind speed of 6.5 m/s. The root-mean-square error (RMSE) between the measured and simulated normalized minimum wind speed is collected for all analyzed turbulence intensity bins in Figure 14. A clear

improvement of the results due to the recalibrated model version up to an ambient turbulence intensity of 16 % is visible. For higher turbulence intensity bins, the RMSE of the recalibrated and the original DWM-Keck model version are similar. The DWM-Egmond model delivers significantly higher RMSEs than the other model versions for all turbulence intensity bins. A comparison between the simulated and measured mean wake wind speed over the rotor area has been carried out as well[1]. The improvement of the mean wind speed is less clear in comparison to the normalized minimum wind speed. Yet, there is

an improvement or results of equal quality are obtained in almost all turbulence intensity bins. At the tails of the wind speed deficit, the curves are coarse, since less scan points are gathered and the influence of turbulence is much higher (see Figure 9). This leads to an error in the mean wake wind speed but not in the minimum wind speed, which is why the illustration and recalibration of the model are based on the minimum wake wind speed instead of the wake mean wind speed.

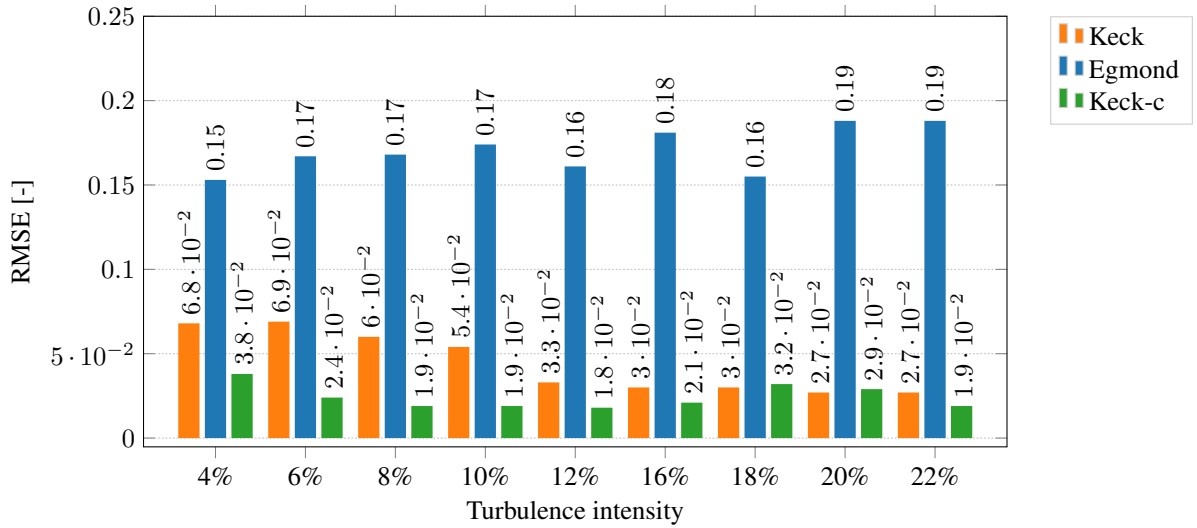

**Figure 14.** RMSE between the LiDAR measured and the simulated normalized minimum wind speed in the wake.

Figure 15 compares the final recalibrated DWM model to the original model definition. It shows the minimum normalized

wind speed (a) and the wind speed at the wake center (b) over downstream distances from $0D$ to $10D$ for the lower and the higher turbulence intensity cases of 6 % and 16 %, respectively. Observing the wind speed at the wake center, higher wind speeds can be seen at lower distances, which derives from the w-shape of the wind speed at these downstream distances. The comparison of the DWM-Keck model (orange curve) and the recalibrated model DWM-Keck-c (green curve) demonstrates that the recalibration leads to a shift of the curve towards lower distances. This shift is more pronounced for the lower turbulence

intensity, leading to a faster degradation of the wind speed deficit. For the higher turbulence intensity, both curves, orange

[1]https://www.wind-energ-sci-discuss.net/wes-2019-89/

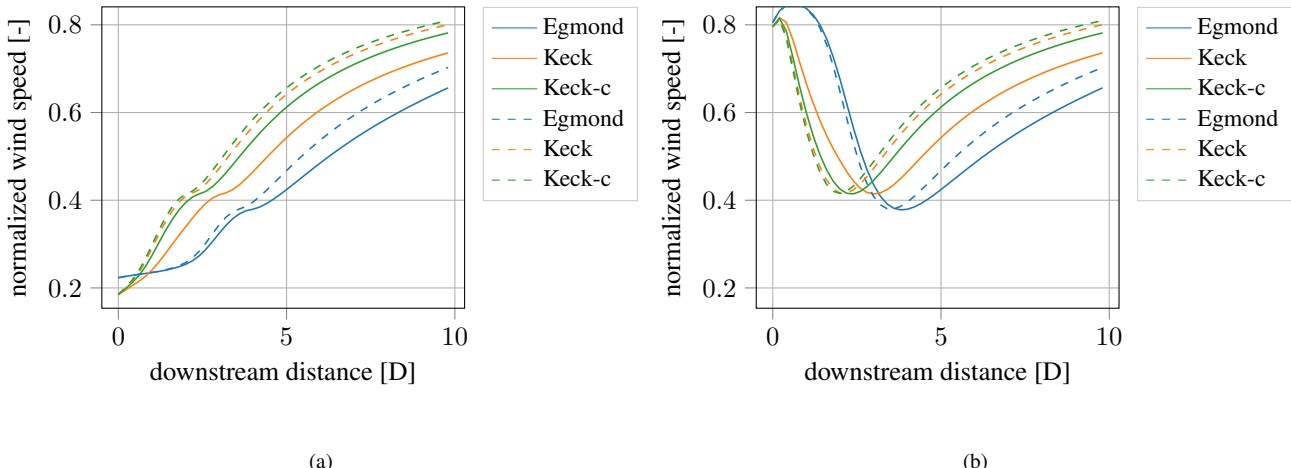

**Figure 15.** Simulated minimal normalized wind speed in the MFR **(a)** and normalized wind speed at the wake center **(b)** over the downstream distance for a turbulence intensity of 6 % (solid curves) and 16 % (dashed curves). The recalibrated model is denoted DWM-Keck-c.

and green, are very close to each other over all distances. The faster degradation of the wind speed deficit in the recalibrated model version is caused by introducing the function $F_{amb}$ in the eddy viscosity definition in Equation (21) as explained in Section 6.3. The function increases the eddy viscosity for lower turbulence intensities and thus increases the wind speed deficit degradation in downstream direction. Contemplating the curve of the minimum wind speed in Figure 15(a), small steps are

formed in the curves between $2D$ and $4D$ (depending on the used model and the turbulence intensity). These steps correspond to the minimum of the curves in Figure 15(b) and are thus related to the transition from the w-shape of the wind speed deficit towards the Gaussian profile and are consequently caused by the resolution in downstream direction. These steps were also found in some measurements and could likewise be related to the implied transition zone.

## 9 Conclusions

The study compares measurements of the wind speed deficit with DWM model simulations. The measurement campaign consists of two nacelle-mounted LiDAR systems in a densely packed onshore wind farm. The LiDAR measurements were prepared by LiDAR and wind field simulations to examine whether the scan pattern is suitable for the outlined analysis. Several wind speed deficits that were simultaneously measured at different downstream distances are presented along with their associated meandering time series. The one-dimensional scan worked reliably in the field campaign, thus, delivering

LiDAR data for a multitude of different ambient conditions. These measurements are compared to the simulated wind speed deficit in the HMFR. The simulation result of the DWM-Keck model is in good agreement, whereas the DWM-Egmond model yields a too low degradation of the wind speed deficit. Furthermore, even the DWM-Keck model shows some discrepancies to the measurements at low turbulence intensities, which is why a recalibrated DWM model was proposed. The recalibrated

model improves the correlation with measurements at low turbulence intensities and leads to an agreement at high turbulence
intensities, which are as good as the original model, thus resulting in a very well overall conformity with the measurements.

Future work will include the analysis of two-dimensional scans as well as measurements with more range gates and higher
spatial resolutions. Increasing the number of range gates and scan points will lead to longer scan times, hence, preventing
further analysis of the wind speed deficit in the MFR and the determination of the meandering time series. Nevertheless,
a validation of the wind speed deficit in the FFR with higher resolutions and more distances seems reasonable to prove the
validity of the outlined calibration also for further distances. Furthermore, the analyzed models will be assessd in load as well as
power production simulations and compared to the particular measurement values from the wind farm. Simulations have shown
that the recalibration of the DWM-Keck model can lead up to 13 % lower loads in the turbulent depending components in cases
with small turbine distances and low turbulence intensities, whereas for higher turbulence intensities (>12 %) the difference
between the original and the recalibrated DWM-Keck model model is less than 5 %. The overall influence of the recalibration
on the power output is low (<2 % for all turbulence intensities). So far, only measured single wakes were presented. Yet, a
brief analysis demonstrated that multiple wakes can also be recorded with the described measurement setup. A future step will
therefore be an analysis of multiple wake situations.

*Code and data availability.*  Access to LiDAR and metmast data as well as the source code used for post-processing the data and simulations
can be requested by the authors.

*Author contributions.*  IR performed all simulations, post-processed and analyzed the measurement data and wrote the paper. LS and DS
gave technical advice in regular discussions and reviewed the paper. PD and MB reviewed the paper and supervised the investigations.

*Competing interests.*  The authors declare that they have no conflict of interest.

*Acknowledgements.*  The content of this paper was developed within the project NEW 4.0 (North German Energy Transition 4.0), which is
funded by the Federal Ministry for Economic Affairs and Energy (BMWI).

 **Appendix A: Measurement results**

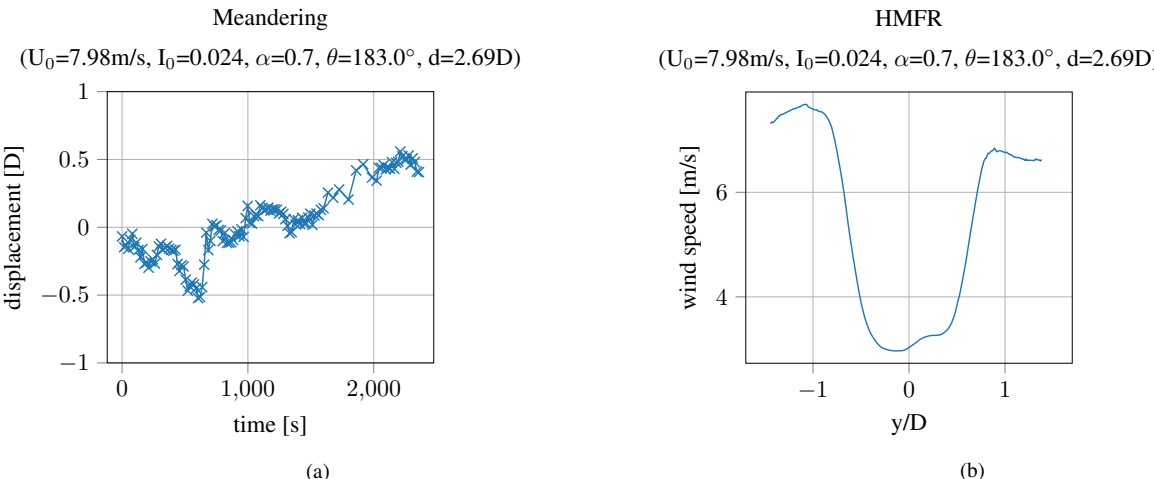

**Figure A1.** Meandering time series **(a)** and wind speed deficit in the HMFR **(b)** at $2.69D$ downstream of the turbine.

**Appendix B: Comparison of measurements and DWM model simulation**

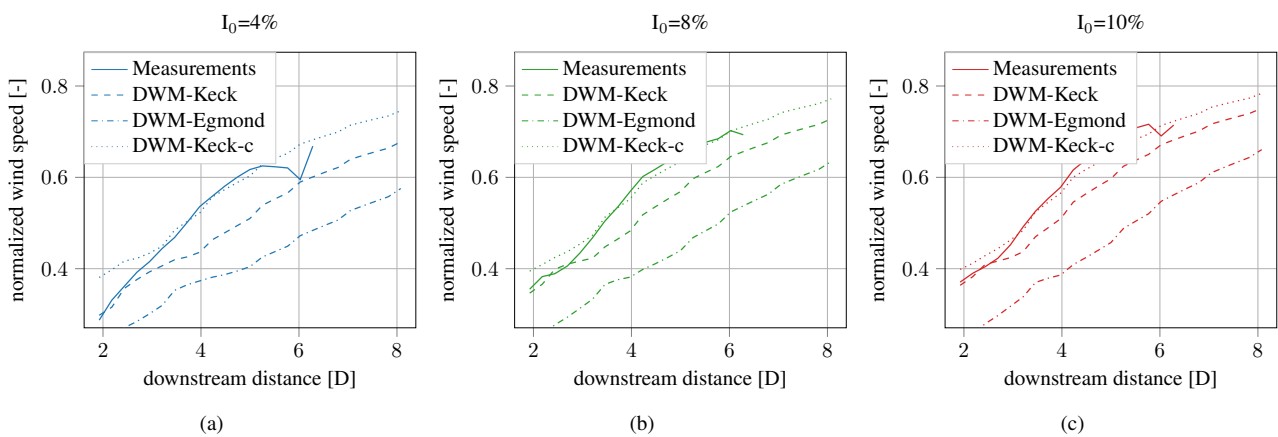

**Figure B1.** Comparison of measurements and simulations of the minimum wind speed deficit in the HMFR for different turbulence intensities. The recalibrated model is denoted DWM-Keck-c.

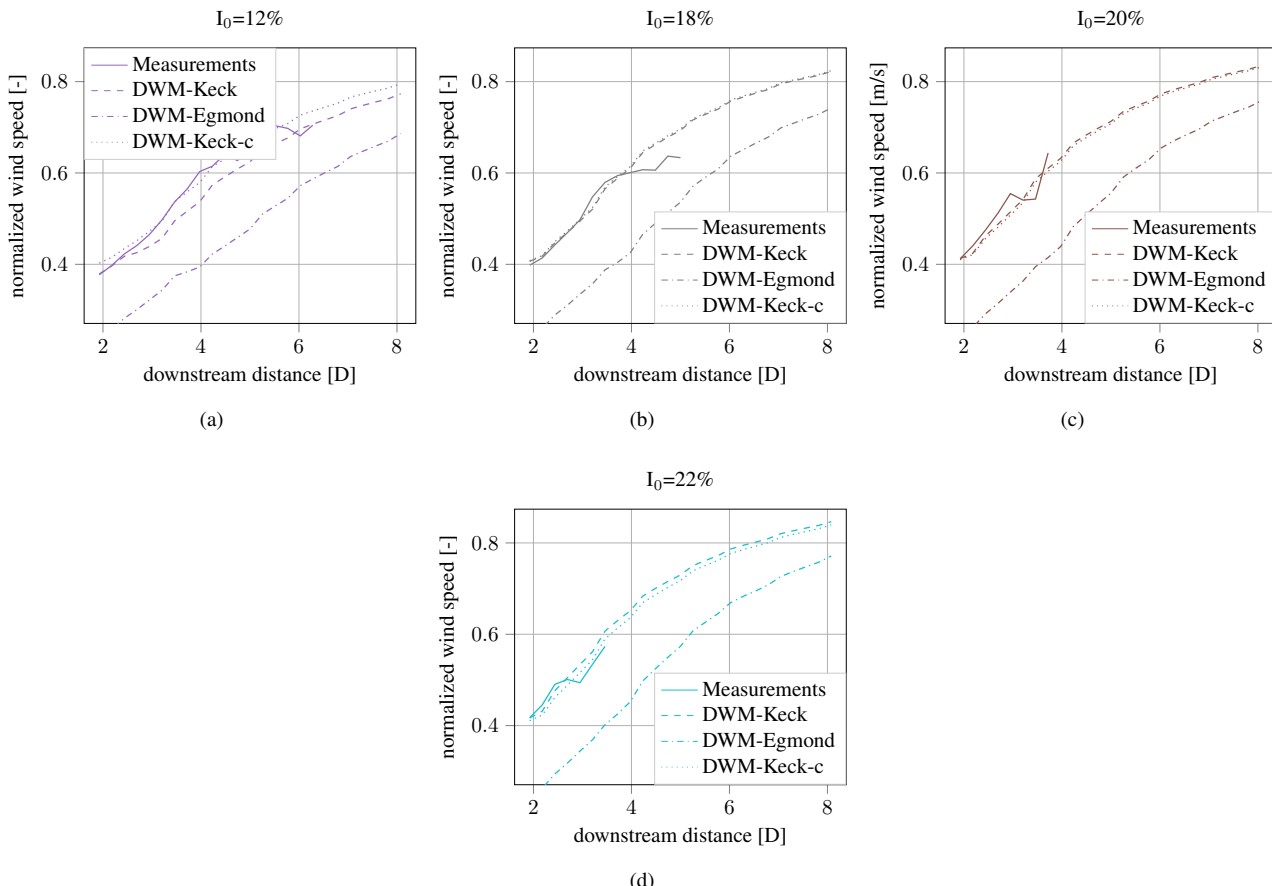

**Figure B2.** Comparison of measurements and simulations of the minimum wind speed deficit in the HMFR for different turbulence intensities. The recalibrated model is denoted DWM-Keck-c.

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
