# Peer review of "DWM model calibration using nacelle-mounted lidar systems"

_Wind Energy Science, 2019_

## Referee Comment (RC1) · Vasilis Pettas (Referee) · 4 Feb 2020

This study presents the results of a measurement campaign, using nacelle mounted lidar pulsed lidars, focusing on the wake of a small, closely-spaced wind farm in a mostly flat terrain. The lidar measurements aim to capture the behavior of the wake in terms of deficit and meandering and compare with numerical predictions of different variations of the dynamic wake meandering (DWM) model. The measurement results are used to calibrate the parameters of one of the variations of the DWM in order to better fit the numerical results to the measurements. The analysis is mostly qualitative.

1. In terms of language the text is well written and clear. Some improvements are needed on the structure and phrasing used. See specific comments.

[Figure]

2. Good literature review especially regarding previous campaigns.

3. The analysis is mainly qualitative. Quantitative analysis should also be used to back up the arguments. More explanation on why we see what we see not only explaining what we see is needed.

4. The data don't seem to be enough especially in higher TIs to generalize the conclusions.

5. Clearer explanation of the tools/methods used especially for the numerical part. How were the virtual lidar measurements done, what software was used, what are the assumptions. . .? This will improve reproducibility and make the work more transparent.

6. More discussion is needed on the uncertainties like yaw direction, sampling rate, low sample size, distance of met mast (especially to WTG2), other wake interactions, LOS reconstruction, etc. How much do all these affect the results? More thorough discussion is needed on these points.

7. Some more tangible results would be good in order for this work to be more useful for the research community. Can the data sets and the codes used to post process and fit the DWM model be made publicly available? Under which condition are the recalibrated parameters valid (terrain, ambient conditions, turbine types etc.)?

8. Language and argumentation need to be more concise. Avoid qualitative terms like "quite", "relatively", "probably", "becomes apparent", "almost" etc. A lot of argumentation is based on oral and visual argumentation instead of quantitative and causal analysis.

The data sets from the measurement campaign and the findings of a recalibrated variation of a specific DWM model can be useful for the wind energy community and is relevant in the context of the journal. Showing the shortcomings of current engineering models as well as issues associated with measurement campaigns has a value for the wind energy research community. I suggest the manuscript for publication after

addressing the comments suggested here.

Please also note the supplement to this comment: https://www.wind-energ-sci-discuss.net/wes-2019-89/wes-2019-89-RC1-supplement.pdf
* * *
[Figure]

**Supplement:**

**Review of "Measuring dynamic wake characteristics with nacelle mounted LiDAR systems" by Inga Reinwardt et al., manuscript number: wes-2019-89**

Vasilis Pettas pettas@ifb.uni-stuttgart.de

**1. General comments**

This study presents the results of a measurement campaign, using nacelle mounted lidar pulsed lidars, focusing on the wake of a small, closely-spaced wind farm in a mostly flat terrain. The lidar measurements aim to capture the behavior of the wake in terms of deficit and meandering and compare with numerical predictions of different variations of the dynamic wake meandering (DWM) model. The measurement results are used to calibrate the parameters of one of the variations of the DWM in order to better fit the numerical results to the measurements. The analysis is mostly qualitative.

- In terms of language the text is well written and clear. Some improvements are needed on the structure and phrasing used. See specific comments.
- Good literature review especially regarding previous campaigns.
- The analysis is mainly qualitative. Quantitative analysis should also be used to back up the arguments. More explanation on why we see what we see not only explaining what we see is needed.
- The data don't seem to be enough especially in higher TIs to generalize the conclusions.
- Clearer explanation of the tools/methods used especially for the numerical part. How were the virtual lidar measurements done, what software was used, what are the assumptions…? This will improve reproducibility and make the work more transparent.
- More discussion is needed on the uncertainties like yaw direction, sampling rate, low sample size, distance of met mast (especially to WTG2), other wake interactions, LOS reconstruction, etc. How much do all these affect the results? More thorough discussion is needed on these points.
- Some more tangible results would be good in order for this work to be more useful for the research community. Can the data sets and the codes used to post process and fit the DWM model be made publicly available? Under which condition are the recalibrated parameters valid (terrain, ambient conditions, turbine types etc.)?
- Language and argumentation need to be more concise. Avoid qualitative terms like "quite", "relatively", "probably", "becomes apparent", "almost" etc. A lot of argumentation is based on oral and visual argumentation instead of quantitative and causal analysis.

The data sets from the measurement campaign and the findings of a recalibrated variation of a specific DWM model can be useful for the wind energy community and is relevant in the context of the journal. Showing the shortcomings of current engineering models as well as issues associated with measurement campaigns has a value for the wind energy research community. I suggest the manuscript for publication after addressing the comments suggested here.

**2. Specific comments**

**Title**

Is the title reflecting the content? Maybe use a more descriptive one, like for instance: "Calibrating a DWM model with measurements of dynamic wake characteristics using nacelle mounted lidar systems"?

**Abstract**

Specify the objectives of the paper clearly, not only the campaign. What do we learn by reading the paper?

I would suggest removing lines 4-5 as no discussion on the optimization procedure is done in the paper itself.

**Introduction**

L 13: Engineering models like the Frandsen model are intended to calculate mainly the wake deficit and shape and not the wake induced turbulence. It should be clearly stated.

L 24: What is meant by 2D wind field here? The lidar can measure 1D (LOS direction only) and 3D in terms of space (have the pulsed technology with range gates). Please clarify.

At the end of the introduction a paragraph should be added stating clearly the objectives of the paper. A small reference on the content and structure of the following sections could also make the work easier to follow.

**Wind farm**

L 67-68: As I understand load measurements are not used in the study, what is the relevance of mentioning the load sensors here?

L 69: LiDAR system of WTG 1 is installed inside the nacelle and measures through a hole in the rear wall: This is an interesting and uncommon setup. Are there any limitations or benefits using this set up? It could be useful information for future campaigns.

L 70: A nacelle mounted GPS is mentioned for the nacelle yaw position tracking. What is the uncertainty of such a system? Did you correlate it with the high frequency SCADA data for nacelle direction? Is there any data filtering based on SCADA or gps nacelle position in order to make sure that the turbines were not yawing often during the accepted time intervals?

L 78-80: The Richardson number is mentioned here. It is not mentioned how it was used to filter the data or how it is used in general in the study. Stability is only mentioned again in L 216 where it is stated that it is not considered. Am I missing something?

L 79: State the heights of measurements used for the calculation of the Richardson number.

How many rays are used in each pulse for the campaign?

Are the SCADA data used 10min averages or high frequency?

How is the met-mast equipped (heights of measurements, measurement devices…)?

L 73-88: A lot of information in this paragraph. Would be clearer to add a table with all the filtering as well as the amount of total data and data kept after filtering. This way it will be easier to identify sources of bad measurements and provide a condensed overview.

L 85-88: Give more details on the final setup of the lidar campaign. What was the sampling rate per scan/ray? Which range gates were used (as 750m exceeds the distance of the downwind turbines and usually this type of devices cannot measure below 50-100m)? Exact information on the campaign can be very useful for future research.

L89-94: Is there any uncertainty in that? According to the misalignment of the nacelle to the main direction, the tilt or yaw flows and the lidar angle, the uncertainty can be significant. Is there a way to quantify that? What are the angles used and how small are they?

The previous comments are correlated to the general comments on discussion of uncertainty and reproducibility of the study.

**Wind speed deficit in MFR calculation**

L 107-108: "However… campaign" This is a good example of more concise language and argumentation needed in the paper. What does highly improbable mean (especially when only 1 10min data set is used for some TI bins later on)? What does very robust mean? Please be more specific in the arguments used to validate assumptions.

L 118-120: Maybe I am missing something, but it is not clear to me how this plausibility check works. Can you explain it more?

**Lidar simulation**

This section needs a lot of work, with a lot of missing information. More information is needed in order to ensure reproducibility. How is the lidar simulator working? How are the wind fields created and how is the DWM model incorporated? Are you using Turbsim or Mannbox generator or some other turbulence generator? How is the LOS speed reconstruction done? Do you consider perfect lidar measurements? How are the range gates and probe volume averaging considered?

L 129-131: This is the only reference through the paper to the optimization study to find an optimal pattern. It results to a simple horizontal scan of 11 equispaced points. It is very general and does not explain the procedure. I think it should either omitted from the paper and only state the used trajectory or add a dedicated section with more details and figures.

L 136: What does very well mean in this context? Can it be quantified e.g. with error metrics or $R^2$?

L 144-146: What does optimal operating conditions mean? Does it mean it operated on max CP, CT which in turn produce the highest deficit? Please be more specific. Maybe a dimensionless CP-CT vs wind speed curve would be useful here but also for the argumentation in L 299 about thrust being constant.

Some references on studies using virtual lidar measurements:

Dimitrov, N., and Natarajan, A., "Application of simulated lidar scanning patterns to constrained Gaussian turbulence fields for load validation," Wind Energy, Vol. 20, No. 1, 2017, pp. 79–95. doi:10.1002/we.1992.

Pettas, V., Costa García, F., Kretschmer, M., Rinker, J. M., Clifton, A., and Cheng, P. W., "A numerical framework for constraining synthetic wind fields with lidar measurements for improved load simulations," AIAA Scitech 2020 Forum, American Institute of Aeronautics and Astronautics, Reston, Virginia, 2020, pp. 1–6. doi:10.2514/6.2020-0993

**Dynamic wake meandering model**

Sections 6.1 and 6.2: Nice, thorough description of the models. Can you explain how you generated the wind fields (tools, models, parameters, discretization) and how you implemented the variations of the DWM models? Is this an in-house tool or a commercial/open source tool? Can the codes be shared with others so that such validations can be repeated with other data sets?

Section 6.3: As stated, the wake induced turbulence in the DWM model is not used in this study. I suggest to remove this section as it does not add something to the purpose of the paper.

L 258-259: What does relatively good agreement mean in this context? Please be more precise and avoid using such expressions.

**Measurement results**

Discussion in this section needs to be more quantitative and not only qualitative and the language needs to be more concise (look at previous comments). It feels more like describing the plots instead of explaining and quantifying what we see (e.g.: "The degradation of the wind speed deficit in downstream direction is clearly identifiable", "The reason is probably the wake of other turbines in the wind farm.", "it becomes apparent that the wind speed deficit in the FFR is less pronounced.", "The most obvious explanation is the reduced number of measurement results in these bins and the higher uncertainty that comes along with it"). I suggest revising this section in terms of phrasing and adding some quantitative results.

The results with high shear and low TI (and vice versa) suggest some kind of stability based filtering in the results. Is this done somehow? Would this be an important parameter on how well the DWM models and the parameter fitting perform?

L 270-271: Can't this (along with the observation that the center of the wake in the MFR is not exactly at the 0 point) correlated to the rotational direction of the rotor too?

L 277-280: This discussion is interesting and would be more relevant if it could quantify the trade-offs. As mentioned in a previous comment this could fit in the numerical study of the optimization.

L 293-298: and L 305 and L 311-314: The data for bins of TI higher than 12 seem very sparse with 1 or 2 data sets each. Are these sufficient to extract conclusions about the models and fit parameters? I would suggest a more thorough argumentation for using them or removing values higher than 12 from the anlysis.

Table 2: Could it include also shear values? Or maybe a plot can be added showing the joint probabilities of shear and TI. This will help to give a better overview of the conditions to the reader.

Figure 2 is hard to read. I recommend plotting it again with thicker lines and playing with line style, markers and size

L298-301: As mentioned earlier, a CP-CT curve vs wind speed would be more clear for this argument.

L 314-318: The argumentation here is weak. More quantitative results are needed and more concise language in order to validate the assumptions.

**Comparison between measurements and DWM model simulation**

L 323: Which are the distances used in the simulations?

L 325-326: "However, the wind speed gradient in axial direction is relatively low and almost linear in the observed downstream distances, so that a fair comparison between simulation and measurements is carried out". The phrasing relatively low and almost linear are not making an argument for the assumptions. Please explain why you consider this valid. Moreover, it is not clear what is meant by fair comparison in this context.

L329 Avoid the phrase 'it is obvious',

L320-336 In general the analysis here is only descriptive and qualitative. Can the convergence be quantified and the discrepancies of the model to the measurements explained based on their assumptions and detail level?

L 342 How were the simulations performed? What code was used, what type of spatial and temporal discretization? Give more details.

L345 It is not clear to me what does this weighting mean. Can you explain it a bit more along with the reasoning?

L 346 It is stated that the calibrated model "coincides very well with the measurements". Can you quantify this improvement by comparing with the level of agreements of the previous models?

L 350-362 In this paragraph the differences between the models described based on figure 10. Can you add some explanation on why the models behave differently? What is the driver of this behavior?

**Conclusions**

L 366 As commented earlier the part about deriving an optimal scan pattern is not discussed at all through the paper. I would suggest you either add a section on this optimization procedure or remove it from the text.

L374 comparably good agreement: This is not clear as a conclusion. As stated earlier I think more concise language and quantitative results are needed.

**3. Minor comments**

L 64 with small turbine distance→five closely spaced Nordex turbines

L 65 in main wind direction → in the main wind direction

L 117 instant in time→ instance of time

L326 donstream→downstream

---

## Referee Comment (RC2) · Helge Aagaard Madsen (Referee) · 25 Feb 2020

**Review of paper wes-2019-89:**

**Measuring dynamic wake characteristics with nacelle mounted LiDAR systems**

by Inga Reinwardt[1], Levin Schilling[1], Peter Dalhoff[1], Dirk Steudel[2], and Michael Breuer[3]

[1]Dep. Mechanical Engineering & Production, HAW Hamburg, Berliner Tor 21, D-20099 Hamburg, Germany

[2]Dep. Turbine Load Calculation, Nordex Energy GmbH, Langenhorner Chaussee 600, D-22419 Hamburg, Germany

[3]Dep. of Fluid Mechanics, Helmut-Schmidt University Hamburg, Holstenhofweg 85, D-22043 Hamburg, Germany

**Brief summary**

The author's present LIDAR measurements of the wake flow behind two different turbines (a 3 MW N117 turbine and a 2.4 MW N117 turbine) in a small wind farm of 5 closely spaced turbines with individual spacing's between 2.51 and 4.71 D. Measurements were carried out for different inflow conditions with turbulence intensities from 4 to 22%. The processing of the measurements comprises a determination of the wake deficit centre by fitting a Gaussian curve to the 11 measured wind speeds. With the deficit centre located the measured points are moved to the meandering frame of reference (MFR), interpolated to a grid and then the mean deficit based on all the individual scans, e.g. 37,  for a 10 min time series is determined.
A comparison between the modelled wake degradation in the MFR and the measured is then conducted. The simulations are carried out with two different versions of the Dynamic Wake Meandering (DWM) model. These versions differ only in the description of the quasi-steady wake deficit. Based on the findings from the LiDAR measurements the formulation of the quasi-steady wake deficit in the DWMmodel was adjusted, so that the recalibrated model coincides very well with the measurements.

**Overall comments**

The subject of the paper on full-scale measurements of the wake characteristics in the fixed frame of reference (FMR) as well as in the meandering frame of reference (MFR) is of considerable importance for the research community as this is the basis for improving our insight into wake flow physics and providing input for improving our modelling capability of wakes. This is important for a more precise prediction of the AEP losses in wind farms as well as of the increased turbine loading.

However, the measurement task is also quite complicated, e.g. due to the considerable size of the wake of a MW turbine and due to the atmospheric flow conditions which means that the wake deficit is masked by this turbulent flow.

The discussion and consideration of this complexity of the measurement task and in particular how it affects the measured wake characteristics should be improved in the final version of the paper. This is important for the use of the results in the paper by the research community. In general, also a more

precise description of approaches, experimental set-up and interpretation of results should be contained in the final version of the paper.

**Specific comments**

**Abstract**

The sentence: "the formulation of the quasi-steady wake deficit in the DWM model has been adjusted" is not precise.

It´s proposed to describe that it´s the correlation of the impact of ambient turbulent to the eddy viscosity that has been investigated and that an improved correlation function (parameter) has been determined based on the present measurements.

**1. Introduction**

Satisfactory description of previous work within the field and the intro to the contents of the present paper at the end.

**2. Wind farm**

Line 68:

- What is the instrumentation in the met. mast ? Please describe in the paper.

Line 76:

- What type of load measurements and have they been used for DWM simulations on the turbines?

**3. Data filtering and processing**

Line 86:

- … "and sorted in accordance with ambient wind speed, ambient turbulence intensity, windshear, atmospheric stability, and wind direction".
  - is it 10 min, mean values that the data are sorted on basis of ?

**4. Wind speed deficit in MFR calculation**

Line 119:

- … "In the analysis presented here only results from a horizontal line scan are analyzed, so that no vertical meandering is considered and the measurement results are fitted to a one-dimensional Gaussian curve defined as follows:"
  - In my view this is an important limitation of the experimental set-up. Overall the impact is that the depth or strength of the deficits are smaller than if the 3D location of the deficits was used. The impact can be investigated using a DWM model and simply set the vertical meandering

to zero. Please discuss this limitation of the measurement set-up and what impact it has on the final result.

Line 137:

- … "After averaging, the plausibility of the results is inspected. If the calculated minimum mean wind speed in the MFR is higher than the minimum mean wind speed in the FFR, it is assumed that the Gauss fit failed and the results are no longer considered.
  - o Besides this plausibility check I would propose to show the standard deviation of all the measurement points around the average MFR from the individual scans, just for a few cases. This will give information about how much averaging is behind the final MFR deficits.

Line 148:

- In figure 2 as I understand the procedure:
  - o – shouldn´t the x axis after the interpolation be in y/d units and not in deg. ?. Likewise in Figure 3b.

**5 LiDAR simulation**

Line 159:

- Were the lidar simulations with the DWM model shown in Figure 3 carried out with ambient turbulence or only a meandering turbulence – please specify?

Line 160:

- … "Whenever the wind speed deficit is mentioned in subsequent validations, it implies the neglection of the vertical meandering, which has only a marginal impact on the shape of the wind speed deficit in the FFR.".
  - o As the meandering turbulence components scales with 0.8 and 0.5 in horizontal and vertical direction relatively to the streamwise turbulence component I am not convinced that this statement is correct. Please expand on this eventually based on simulations with the DWM model.

**6 Dynamic wake meandering model**

Line 175:

- … "It compares directly to the LiDAR measurements after transforming the measurements into the MFR as explained in the last section".
  - o As mentioned above the measured wake deficit might be less sharp (deep) due to neglecting the vertical meandering and due to the averaging of many individual deficits impacted by ambient turbulence.

Line 189:

- … "The error that inherently comes with this assumption is accommodated by using the wind speed deficit two rotor diameters downstream (beginning of the far-wake area) as a boundary condition for the solution of the thin shear-layer equations. "
  - o It might be important to point out here that the eddy viscosity model in the DTU DWM implementation is run from the rotor plane and downstream with the fully expanded wake deficit (eq. 6 and 7) as boundary conditions but where a fit of the deficit at 2D downstream to Actuator Disc simulations determined eq. 8 and the filter function for non- turbulent flow.

Line 272:

- … "It shows that for lower turbulence intensities and moderate to high turbine distances the wind speed deficit degradation is too low."
  - o Maybe write "was too low in the model version from 2010 – ref J. Sol. Energy Eng., 132, 041 014, 2010. " The deviations were the reason to recalibrate the model as presented in the 2013 paper.

**7 Measurement results**

Line 289:

- … "The corresponding mean wind speed deficit is illustrated in Figure 6(b)."
  - o In order to evaluate what this mean deficit it would be valuable if the standard deviation of the 11 raw measurement points for each scan are shown

Line 310:

- … "The reason is probably the wake of other turbines in the wind farm".
  - o It could also be due to wake rotation as seen in 3D CFD rotor simulations in sheared inflow. It shows that high velocity flow at one side of the rotor is rotated down towards the ground and the opposite on the other side of the turbine.

Line 323:

- … "In this range both turbines operate under optimal and most efficient conditions resulting in maximum energy output from the wind. The thrust coefficient is constant in this region. Therefore, the axial induction and the wind speed deficit normalized by the turbine's inflow wind speed are also expected to be constant for similar ambient conditions over this wind speed range."
  - o Its mentioned ".. expected to be constant". What is actually used in the DWM simulations ?

- o Further down at line 368 is mentioned : ".. that the axial induction of both turbines is slightly different under partial load conditions." So is the detailed aero loading of each of the two turbines are simulated ?

**8 Comparison between measurements and DWM model simulation**

Line 358:

- … "For lower turbulence intensities and higher distances (greater than 3D) there is a relatively large discrepancy between measurements and simulations. A similar observation was made in Larsen et al. (2013)."
    - o This comment was on the model before the recalibration so it should be deleted if pointing to the "DWM-Egmond model"

Line 362

- As concerns the results in Figure 10 and Figure 11 for the DWM-Egmond model they seem not to agree with simulations with our DTU implementation of the DWM model, however with the uncertainty of just assuming a similar turbine operation but without knowing the details of the turbine
    - o The authors are encouraged to share and upload more details of their simulations so that the results can be checked with an original implementation of the so-called DWM-Egmond model.
    - o Further, it is proposed to show a figure with e.g. the mean velocity of the wake deficit or the mean velocity cubed (to show reduction in power of the downstream turbine) and otherwise in the same way as Figure 9. The mean velocity is a more robust characterization of the wake deficit than the minimum value velocity within the deficit. The minimum value can easily be influence by the details of the aerodynamic modelling of the turbine.

**Some final conclusive remarks**

- There is no discussing of the impact of the findings. Changing the wake recovery characteristics have obviously an impact on power production and loads.
    - o For the Egmond aan Zee case the DWM model was as mentioned calibrated to the power reduction of the second turbine in a row relative to the first one for different spacings and turbulence intensities. Using this calibration an overall good correlation of simulated and measured loads was found.
    - o Have the present recalibrated model been used for power and load simulations and compared with measurements in the present wind farm?
- The reviewer finds that due to the above mentioned uncertainties/limitations related to the measurements of the deficits in the meandering frame of reference there will be a bias of the measured deficits being more smooth. Please comment on this view.

**Final conclusion of review**

The reviewer can recommend publication of the paper considering the above comments/questions and not least work together on a check of the DWM-egmond simulations by comparing with the DTU implementation of the DWM model as this is the basis for the use of the DWM model in certification.

(IEC 61400-1 Ed.4: IEC 61400-1 Ed. 4: Wind energy generation systems - Part 1: Design requirements, Guideline, International Electrotechnical Commission (IEC), 2019.

---

## Author Comment (AC1) · 25 Mar 2020

**Responses to the referee:** Vasilis Pettas (pettas@ifb.uni-stuttgart.de)
 Received and published:  04.02.2020

**1. General comments**
Thank you very much for the detailed and helpful comments on the paper. We appreciate the work of reviewing it and believe that the quality of the paper significantly gained from the comments.

In the following, we respond to all comments in detail. Particular focus is given to the main comment on more quantitative results and their precise description by introducing further graphs and explanations. The uncertainties are discussed more in detail as well as the data filtering procedure.

**2. Specific comments**
**Title**
Is the title reflecting the content? Maybe use a more descriptive one, like for instance: "Calibrating a DWM model with measurements of dynamic wake characteristics using nacelle mounted lidar systems"?
Response: A more precise title is reasonable and adjusted to: DWM model calibration using nacelle mounted lidar systems.

**Abstract**
Specify the objectives of the paper clearly, not only the campaign.
What do we learn by reading the paper?
I would suggest removing lines 4 - 5 as no discussion on the optimization procedure is done in the paper itself.
Response: Line 4-5 were removed.

**Introduction**
L 13: Engineering models like the Frandsen model are intended to calculate mainly the wake deficit and shape and not the wake induced turbulence. It should be clearly stated.
Response: As far as we know, the Frandsen model is commonly used in the industry to calculate the wake induced turbulence and is also recommended in the IEC guideline. What kind of model do you mean here?

L 24: What is meant by 2D wind field here?
The lidar can measure 1D (LOS direction only) and 3D in terms of space (have the pulsed technology with range gates). Please clarify.
Response: The text passage was rephrased to: *"Especially, the so-called scanning LiDAR systems offer great potential for detailed wake analysis. These LiDARs are capable of scanning a three-dimensional wind field, so that the line of sight (LOS) wind speed can be measured subsequently at different positions in the wake, thus enabling the detection of the wake meandering as well as the shape of the wind speed deficit in the MFR."*

At the end of the introduction a paragraph should be added stating clearly the objectives of the paper. A small reference on the content and structure of the following sections could also make the work easier to follow.

Response: A clear outline of the objectives was added and a small overview of the structure of the following sections was given:

*"Thus, a detailed comparison of the predicted degradation of the wind speed deficit between the DWM model and the measurement results is possible. Furthermore, the collected LiDAR measurements are used to recalibrate the DWM model, so that the wake degradation can be modeled more precisely. As a consequence, the calculation of the loads as well as the energy yield of the wind farm can be improved. The remaining document is arranged as follows: in Section 2, the investigated wind farm and the installed measurement equipment are described in detail. Afterwards, in Section 3, an explanation of the data processing and filtering of the measurement results is given. Sections 4, 5, and 6 focus on the description of the theoretical background and a hands-on implementation of the DWM model is introduced. Based on the outlined measurement results, a recalibration of the defined degradation of the wind speed deficit in the DWM model is proposed in Section 6. A summary of the measurement results can be found in Section 7 and a comparison to the original DWM model as well as the recalibrated version is presented in Section 8. Eventually, all findings are concluded in Section 9."*

**Wind farm**

L 67-68: As I understand load measurements are not used in the study, what is the relevance of mentioning the load sensors here

Response: The load measurements should be used in a subsequent publication and should be used to further verify the recalibration of the DWM model. A hint to future objectives is added.

L 69: LiDAR system of WTG 1 is installed inside the nacelle and measures through a hole in the rear wall: This is an interesting and uncommon setup. Are there any limitations or benefits using this set up? It could be useful information for future campaign

Response: Mounting the device on top of the nacelle of WTG 1 is not possible, as the area is occupied by a recuperator. The reason and the limitations that accompany it are complemented.

L 70: A nacelle mounted GPS is mentioned for the nacelle yaw position tracking. What is the uncertainty of such a system? Did you correlate it with the high frequency SCADA data for nacelle direction?
Is there any data filtering based on SCADA or gps nacelle position in order to make sure that the turbines were not yawing often during the accepted time intervals?

Response: The differential GPS system measures in centimetre range. A comparison between the SCADA data nacelle direction has been done. This comparison has shown that the measured nacelle direction of the SCADA system has a non-negligible offset of more than 10° at some turbines. This error in measuring the nacelle direction occurs frequently at common wind turbines, wherefore we decided to install the GPS systems. The GPS systems are used to ensure that the turbines are not yawing during the used time intervals. This is also mentioned in Section 4.

L 78 - 80: The Richardson number is mentioned here. It is not mentioned how it was used to filter the data or how it is used in general in the study.
Stability is only mentioned again in L 216 where it is stated that it is not considered. Am I missing something?
L 79:State the heights of measurements used for the calculation of the Richardson number
Response: First, we divided the data set into stable and instable measurements, but in the end, we used all data sets for the recalibration of the DWM model. Therefore, the Richardson number calculation was removed. Furthermore, in L216, in the description of the DWM-Keck model, it is explicitly pointed out that no atmospheric stability is included in the model, as the referred author of this model version developed a model with atmospheric stability included and to clarify that this approach is not used here.

How many rays are used in each pulse for the campaign?
Response: The pulse repetition rate of the LiDAR system is 15 kHz and the ray update rate is about 1Hz, so it averages over approximately 15,000 pulses (depending on the atmospheric conditions). The sample frequency is 100 MHz. Considering the speed of light, we get a point length of 1.5 m. The range gate length is 30 m, thus 20 points are used per range gate. This explanation was also added to the paper.

Are the SCADA data used 10min averages or high frequency?
Response: The SCADA data is only used to determine if the turbine operates under normal power production conditions and to affirm no yaw misalignment. For this purpose, the statistics of the 10-min time series are used. All other data filtering is done with the metmast and the GPS systems.

L 73-88: A lot of information in this paragraph. Would be clearer to add a table with all the filtering as well as the amount of total data and data kept after filtering. This way it will be easier to identify sources of bad measurements and provide a condensed overview.
Response: The paragraph was restructured, and a workflow was added to clarify the filtering procedure. The amount of data after filtering has already been given in Table 2 in the results section.

L 85-88: Give more details on the final setup of the lidar campaign. What was the sampling rate per scan/ray? Which range gates were used (as 750m exceeds the distance of the downwind turbines and usually this type of devices cannot measure below 50-100m)? Exact information on the campaign can be very useful for future research.
Response: The sampling rate as well as the range gates were added in the section (see also previous comment). The range gates used for the validation of the DWM model and the recalibration can vary between each used time series because not all range gates fulfill the filtering criteria. Nevertheless, the used range gates for all data sets are illustrated in Figures 6, 7, and 8. Further distances, which are not illustrated in these graphs, are not considered.

L89-94: Is there any uncertainty in that? According to the misalignment of the nacelle to the main direction, the tilt or yaw flows and the lidar angle, the uncertainty can be significant. Is there a way to quantify that? What are the angles used and how small are they?
Response: A discussion and estimation of the error made by yaw misalignments was supplemented as follows: *"…if there is yaw misalignment, this could have an impact on the overall results. To decrease the uncertainties based on yaw misalignments, the measurement*

*data has accordingly been filtered. The yaw misalignment has the biggest impact at the largest scan opening angles, so that a misalignment of 6° at an opening angle of 20° leads to an overestimation of the wind speed of less than 5%.".*

**Wind speed deficit in MFR calculation**
L 107-108: "However… campaign" This is a good example of more concise language and argumentation needed in the paper. What does highly improbable mean (especially when only 1 10min data set is used for some TI bins later on)? What does very robust mean? Please be more specific in the arguments used to validate assumptions.
Response: Results of the calculation of the position of the wind speed deficit at 200 m based on the DWM model simulation has been added to clarify the very low probability ("*e.g., the DWM model predicts the wind speed deficit's probability at the horizontal position of 200 m to be $2*10^{-22}$ for an ambient wind speed of 6.5 m/s and an ambient turbulence intensity of 8 %*").
What is meant by "especially when only 1 10min data set is used for some TI bins later on"? Are you suggesting that too much data is filtered out due to the 200 m criterion? Based on the simulation results given from the DWM model, this is not the case.

L 118-120: Maybe I am missing something, but it is not clear to me how this plausibility check works. Can you explain it more?
Response: After averaging the wind speed deficits in the MFR and FFR, the calculated minimum mean wind speed in the MFR is compared to the minimum mean wind speed in the FFR. In theory, the wind speed deficit in the MFR should be more pronounced than the measured one in the FFR. This comparison is used as a plausibility check.

**Lidar simulation**
This section needs a lot of work, with a lot of missing information. More information is needed in order to ensure reproducibility. How is the lidar simulator working? How are the wind fields created and how is the DWM model incorporated? Are you using Turbsim or Mannbox generator or some other turbulence generator? How is the LOS speed reconstruction done? Do you consider perfect lidar measurements? How are the range gates and probe volume averaging considered?
Response: The LiDAR simulations are very simple and basic to ensure that the meandering as well as the wind speed deficit in the MFR could be captured with the used devices and to check if the selected scan pattern is usable. The wind field with wake effects is generated with an in-house Python tool. A detailed description of the model implementation is given in Section 6 and is not repeated here. A hint to the next section has already been given. There, it is explained that the Veers model is used instead of the Mannbox. The simulations assume perfect LiDAR measurements, so that no probe volume averaging is considered and the LiDAR directly measures the horizontal wind speed. The wind field is simulated at midway of the range gate.

L 129-131: This is the only reference through the paper to the optimization study to find an optimal pattern. It results to a simple horizontal scan of 11 equispaced points. It is very general and does not explain the procedure. I think it should either omitted from the paper and only state the used trajectory or add a dedicated section with more details and figures.
Response: L129-131 has been rephrased. The LiDAR simulation are only used to check if the scan pattern could be used in the campaign and only manual iteration processes with

different angle increments have been carried out. To avoid further misunderstandings, the term "optimization" has been replaced in the description. Graphs with results of simulation and simulated "measurements" are given in Figure 3.

L 136: What does very well mean in this context? Can it be quantified e.g. with error metrics or R^2?
Response: The coefficient of determination is given in Figure 3 ($R^2$=0.93). A hint in the text was added, too.

L 144-146: What does optimal operating conditions mean? Does it mean it operated on max CP, CT which in turn produce the highest deficit? Please be more specific. Maybe a dimensionless CP-CT vs wind speed curve would be useful here but also for the argumentation in L 299 about thrust being constant.
Response: Yes, optimal operating conditions means operating at maximum CP, where the highest or most pronounced deficit is generated. CP and CT curves are added in the section "wind farm".

**Dynamic wake meandering model**
Sections 6.1 and 6.2: Nice, thorough description of the models. Can you explain how you generated the wind fields (tools, models, parameters, discretization) and how you implemented the variations of the DWM models? Is this an in-house tool or a commercial/open source tool? Can the codes be shared with others so that such validations can be repeated with other data sets?
Response: The wind fields are generated with an in-house Python tool, as mentioned in Section 6 and described in Section 6.1. The discretization in axial and radial direction for solving the thin shear layer equations is 0.2D and 0.0125. The information was added to the description. The axial induction factor, which is needed for calculating the boundary conditions, cannot be shared because these are confidential data of the turbine manufacturer. All other parameters are given. The source code can be requested by the authors as explained at the end of the paper in the provided section "Code and data availability".

Section 6.3: As stated, the wake induced turbulence in the DWM model is not used in this study. I suggest to remove this section as it does not add something to the purpose of the paper.
Response: The section was removed.

L 258-259: What does relatively good agreement mean in this context? Please be more precise and avoid using such expressions.
Response: The sentence was rephrased.

The results with high shear and low TI (and vice versa) suggest some kind of stability based filtering in the results. Is this done somehow? Would this be an important parameter on how well the DWM models and the parameter fitting perform?
Response: There is no stability filtering included in the results. Previously, a filtering according to atmospheric stability was implemented, but since this drastically decreases the amount of data, it was discarded and only a sorting according to turbulence intensity bins has been carried out. Moreover, the used eddy viscosity description in the DWM model,

which is calibrated, only depends on the turbulence intensity, thus atmospheric stability is only partially and indirectly considered in the model description, which is why a classification into turbulence intensity bins is more valuable in this application.

L 270-271: Can't this (along with the observation that the center of the wake in the MFR is not exactly at the 0 point) correlated to the rotational direction of the rotor too?
Response: The movement of the wake is based on the assumption that the wake behaves as a passive tracer in a turbulent ambient wind field, so the movement is driven by large scale turbulences and not by the rotational direction of the rotor. Furthermore, if the displacement would be correlated to the rotational direction of the rotor, this behaviour should be visible in all data sets, which is not the case.

L 277-280: This discussion is interesting and would be more relevant if it could quantify the trade-offs. As mentioned in a previous comment this could fit in the numerical study of the optimization.
Response: A quantitative discussion of the possibility of increasing the number of scan points was added: *"According to Equation (18), the meandering is correlated to frequencies lower than approximately 0.028 Hz considering a wind speed of 6.5 m/s and a rotor diameter of 117 m. This means that, considering the Nyquist–Shannon sampling theorem, the scan time must be longer than half of the reciprocal of 0.028 Hz, which results in a necessary scan time of less than 18 s. The scan time for the current usage of 11 scan points is already at about 16 s (depending on the visibility conditions), which is close to the limit of 18 s, so with an increased number of scan points it is no longer ensured that the meandering can be captured."*

L 293-298: and L 305 and L 311-314: The data for bins of TI higher than 12 seem very sparse with 1 or 2 data sets each. Are these sufficient to extract conclusions about the models and fit parameters? I would suggest a more thorough argumentation for using them or removing values higher than 12 from the analysis.
Response: The agreement between the measurements and the simulations is already good in the higher turbulence intensity bins, so the recalibration affects only the lower turbulence intensity bins with larger amounts of data, while the influence of the calibration on higher turbulence intensities is negligible. Therefore, it would not make any difference to exclude the data from the model fit. This explanation is added at the end of Section 7.

Table 2: Could it include also shear values? Or maybe a plot can be added showing the joint probabilities of shear and TI. This will help to give a better overview of the conditions to the reader.
Response: A scatterplot of shear and TI was added.

Figure 2 is hard to read. I recommend plotting it again with thicker lines and playing with line style, markers and size
Response: The authors think that the method description in Figure 2 is sufficient.

L298-301: As mentioned earlier, a CP-CT curve vs wind speed would be more clear for this argument.
Response: CP-CT curves were added and referred to.

L 314-318: The argumentation here is weak. More quantitative results are needed and more concise language in order to validate the assumptions.

Response: A more detailed description about the uncertainties related to the determination of the ambient conditions as well as a description, why it is acceptable to use the higher turbulence intensity bins for the recalibration (see also comment to line L 293-298: and L 305 and L 311-314 above), was added as follows: *"The farthest distance between the metmast and the measured wind speed with the LiDAR system, which can occur in the analyzed sectors, is about 1200 m. With an ambient wind speed of 6.5 m/s, this leads to a wake advection time of 185 s, thus even at worst conditions, the measured ambient conditions at the metmast should be valid for the measured wakes from the LiDAR system most of the time. Furthermore, there is no complex terrain at the site, so it can be assumed that the conditions do not change with the wind direction. In addition, the agreement between measurements and simulations is already good in the higher turbulence intensity bins, so the recalibration affects only the lower turbulence intensity bins with larger amounts of data, while the influence of the calibration on higher turbulence intensities is negligible (see Figure 14)."*

**Comparison between measurements and DWM model simulation**

L 323: Which are the distances used in the simulations?

Response: As explained, the simulated distances correspond to the center of the range gate.

L 325-326: "However, the wind speed gradient in axial direction is relatively low and almost linear in the observed downstream distances, so that a fair comparison between simulation and measurements is carried out". The phrasing relatively low and almost linear are not making an argument for the assumptions. Please explain why you consider this valid. Moreover, it is not clear what is meant by fair comparison in this context.

Response: The following explanation was added: *"The wind speed gradient in axial direction is low and almost linear in the observed downstream distances, so even in the DWM model, the discretization in downstream direction is 23.4 m (equivalent to 0.2D), which is in the same magnitude as the range gate of 30 m. Therefore, a valid comparison between simulation and measurements is carried out."*

L329 Avoid the phrase 'it is obvious',

Response: Phrase was removed.

L320-336 In general the analysis here is only descriptive and qualitative. Can the convergence be quantified and the discrepancies of the model to the measurements explained based on their assumptions and detail level?

Response: A graph with the RMSE between the simulations and models was added as well as a comparison of the deviations to the measurement uncertainties that are related to yaw misalignments and measuring the LOS wind speed itself.

L 342 How were the simulations performed? What code was used, what type of spatial and temporal discretization? Give more details.

Response: A detailed explanation of the simulations is given in Section 6. It is done with an in-house python tool. The spatial and temporal resolution were also added in this section.

L345 It is not clear to me what does this weighting mean. Can you explain it a bit more along with the reasoning?

*Response:* To calculate a mean value of the simulated minimum wind speed and thus allow a comparison with the measurement results collected at two different turbine types, simulations with both turbine types are carried out for each turbulence intensity bin and weighted in accordance with the number of measurement results per turbine listed in Table 2. Thus, for example at the ambient turbulence intensity bin of 4 %, the mean value of the simulated minimum wind speed consists of the sum of the simulated minimum wind speed weighted by 0.451 and 0.549, the weighting factors for WTG1 and WTG2, respectively. Nevertheless, this weighting has only a marginal influence on the overall results, because the axial induction in the considered wind speed range (5 m/s – 8 m/s) is very small for these two turbine types (see also thrust and power curves in Figure 3). A more detailed explanation was also added in the paper.

L 346 It is stated that the calibrated model "coincides very well with the measurements". Can you quantify this improvement by comparing with the level of agreements of the previous models?

*Response:* A graph with the RMSE between measurements and simulations for all turbulence intensity bins was added to provide a better quantification of the improvements.

L 350 - 362 In this paragraph the differences between the models described based on Figure 10. Can you add some explanation on why the models behave differently? What is the driver of this behavior?

*Response:* The difference between the models was explained in detail in Section 6.1 and repeated in Section 8. The DWM-Egmond model and the DWM-Keck model differ in the definition of the boundary conditions for solving the thin shear layer equations as well as the eddy viscosity definition, which is used to calculate the expansion downstream. The DWM-Keck and the recalibrated DWM-Keck-c model differ in the definition of the eddy viscosity. The faster degradation of the wind speed deficit in the recalibrated model version is caused by introducing the function $F_{amb}$ in the eddy viscosity definition in Equation (21) as explained in Section 6.1. The function increases the eddy viscosity for lower turbulence intensities and thus increases the wind speed deficit degradation in downstream direction.

**Conclusions**

L 366 As commented earlier the part about deriving an optimal scan pattern is not discussed at all through the paper. I would suggest you either add a section on this optimization procedure or remove it from the text.

*Response:* The sentence was rephrased.

L374 comparably good agreement: This is not clear as a conclusion. As stated earlier I think more concise language and quantitative results are needed

*Response:* The sentence was rephrased.

**3. Minor comments**

*Response:* All minor comments were adopted in the paper.

---

## Author Comment (AC2) · 25 Mar 2020

**Responses to the referee:** Helge Aagaard Madsen (hama@dtu.dk)
Received and published: 25.02.2020

We are delighted for your valuable comments on the paper. Considering your comments leads to a significant improvement of the paper. We thank you a lot for taking the time to review this paper.

**Specific comments**
**Abstract**

The sentence: "the formulation of the quasi-steady wake deficit in the DWM model has been adjusted" is not precise.
It´s proposed to describe that it´s the correlation of the impact of ambient turbulent to the eddy viscosity that has been investigated and that an improved correlation function (parameter) has been determined based on the present measurements.

Response: The sentence has been adjusted to: *"Based on the findings from the LiDAR measurements, the impact of the ambient turbulence intensity on the eddy viscosity definition in the quasi-steady deficit has been investigated and, subsequently, an improved correlation function has been determined, resulting in very good conformity between the new model and the measurements."*

**2. Wind farm**

Line 68:
- What is the instrumentation in the met mast ? Please describe in the paper.

Response: It is equipped with 11 anemometers, two of which are ultrasonic devices, three wind vanes, two temperature sensors, two hygrometers, and two barometers. The sensors are distributed along the whole metmast, but at least one of each is mounted in the upper eight meters. A Figure with the instrumentation and measurement heights was added.

Line 76:
- What type of load measurements and have they been used for DWM simulations on the turbines?

Response: Strain gauges are installed at the three turbines to measure tower bottom, tower top as well as blade edge- and flapwise moments. Unfortunately, the load measurements are not in the scope of this paper but will be introduced in future publications, i.a., to verify the recalibration. A hint that these load measurements are used for further investigations was added.

**3. Data filtering and processing**

Line 86:

- … "and sorted in accordance with ambient wind speed, ambient turbulence intensity, windshear, atmospheric stability, and wind direction".
  - is it 10 min, mean values that the data are sorted on basis of ?

Response: Yes, the data are filtered based on the 10-min time series statistics from the metmast. The information was added to the manuscript.

**4. Wind speed deficit in MFR calculation**
Line 119:

- … "In the analysis presented here only results from a horizontal line scan are analyzed, so that no vertical meandering is considered and the measurement results are fitted to a one-dimensional Gaussian curve defined as follows:"
  - In my view this is an important limitation of the experimental set-up. Overall the impact is that the depth or strength of the deficits are smaller than if the 3D location of the deficits was used. The impact can be investigated using a DWM model and simply set the vertical meandering to zero. Please discuss this limitation of the measurement set-up and what impact it has on the final result.

Response: A comparison of the simulated wind speed deficit with the DWM model in the complete MFR and without eliminating the vertical meandering in the wind speed deficit was added. There are only small discrepancies around the center of the wake. Nevertheless, in the comparison between the simulated wind speed deficit and the measured wind speed deficit the vertical meandering is not eliminated, so that in both cases the wind speed deficit is similarly reduced in depth. Naturally, the minimum wake wind speed deficit in the MFR without elimination of the vertical meandering is used for the recalibration, too. To clarify that the vertical meandering is not eliminated in any case, but included in the wind speed deficit, the abbreviation HMFR (horizontal meandering frame of reference) is introduced and used instead of MFR.

Line 137:

- … "After averaging, the plausibility of the results is inspected. If the calculated minimum mean wind speed in the MFR is higher than the minimum mean wind speed in the FFR, it is assumed that the Gauss fit failed and the results are no longer considered.
  - Besides this plausibility check I would propose to show the standard deviation of all the measurement points around the average MFR from the individual scans, just for a few cases. This will give information about how much averaging is behind the final MFR deficits.

Response: The plots for the corresponding turbulence intensities for Figure 6 (HMFR) and 7 (FFR) are given below. The comparison of the turbulence intensity in the HMFR and FFR show a decrease of the two maxima at the turbulence intensity in the HMFR, which is expected due to the transformation to the HMFR. The two maxima do not vanish completely in the HMFR graphs due to the small-scale turbulence, which is related to blade tip and root vortices as well as the wake shear itself. Additionally, the turbulence which is related to the vertical meandering is still included. Furthermore, the ambient turbulence intensity of

11.7% and 2.4% can be seen towards the edges of the curve, where the wake influence decreases.

[Figure]

Line 148:
- In figure 2 as I understand the procedure:
  - – shouldn´t the x axis after the interpolation be in y/d units and not in deg. ?. Likewise in Figure 3b.

Response: The label refers to the scan direction, because it is the interpolated scan direction. Nevertheless, it is clearer if the axis is in y/d to correspond to the Figures in section 7. Both graphs were adjusted.

**5. LiDAR simulation**
Line 159:
- Were the lidar simulations with the DWM model shown in Figure 3 carried out with ambient turbulence or only a meandering turbulence – please specify?

Response: It is the complete DWM model wind field with ambient turbulence. It is specified in the text.

Line 160:

- … "Whenever the wind speed deficit is mentioned in subsequent validations, it implies the neglection of the vertical meandering, which has only a marginal impact on the shape of the wind speed deficit in the FFR.".
  - As the meandering turbulence components scales with 0.8 and 0.5 in horizontal and vertical direction relatively to the streamwise turbulence component I am not convinced that this statement is correct. Please expand on this eventually based on simulations with the DWM model.

Response: A comparison of the simulated wind speed deficit with DWM model in the complete MFR and the HMFR was added (see also response to comment on Line 119).

**6. Dynamic wake meandering model**

Line 175:

- … "It compares directly to the LiDAR measurements after transforming the measurements into the MFR as explained in the last section".
  - As mentioned above the measured wake deficit might be less sharp (deep) due to neglecting the vertical meandering and due to the averaging of many individual deficits impacted by ambient turbulence.

Response: That is true, although, the DWM model simulations showed that the influence is small. In the comparison between the simulated and the measured wind speed deficit the vertical meandering is also neglected, hence in both cases the wind speed deficit is less deep. Since the sentence seems to be misleading, it was deleted.

Line 189:

- … "The error that inherently comes with this assumption is accommodated by using the wind speed deficit two rotor diameters downstream (beginning of the far-wake area) as a boundary condition for the solution of the thin shear-layer equations. "
  - It might be important to point out here that the eddy viscosity model in the DTU DWM implementation is run from the rotor plane and downstream with the fully expanded wake deficit (eq. 6 and 7) as boundary conditions but where a fit of the deficit at 2D downstream to Actuator Disc simulations determined eq. 8 and the filter function for non- turbulent flow.

Response: The equations are also directly solved from the rotor plane in the implementation here. It is rephrased to:
*"The error that inherently comes with this assumption is accommodated by using the wind speed deficit two rotor diameters downstream (beginning of the far-wake area) as a boundary condition for the solution of the thin shear layer equations. The equations are solved directly from the rotor plane by a finite-differences method with a discretization in axial and radial direction of 0.2D and 0.0125D combined with an eddy viscosity ($v_T$) closure approach."*
In section 6.1.1 DWM-Egmond following sentence was added:
*"The filter function as well as Equation 8 are calibrated against actuator disc simulations at a downstream distance of 2D, the beginning of the far-wake area, where the wake is fully expanded (Madsen et al., 2010)."*

Line 272:

- … "It shows that for lower turbulence intensities and moderate to high turbine distances the wind speed deficit degradation is too low."
    - o Maybe write "was too low in the model version from 2010 – ref J. Sol. Energy Eng., 132, 041 014, 2010." The deviations were the reason to recalibrate the model as presented in the 2013 paper.

Response: This sentence is rephrased to: *"It shows that the wind speed deficit degradation is too low for lower turbulence intensities and moderate to high turbine distances in the model version from Madsen et al. (2010). For this reason, the downstream distance dependent function $F_{amb}$ was introduced into the eddy viscosity description in Larsen et al. (2013)."*

**7. Measurement results**

Line 289:

- … "The corresponding mean wind speed deficit is illustrated in Figure 6(b)."
    - o In order to evaluate what this mean deficit it would be valuable if the standard deviation of the 11 raw measurement points for each scan are shown

Response: The plots of the corresponding turbulence intensities are given in the comment on Line 137.

Line 310:

- … "The reason is probably the wake of other turbines in the wind farm".
    - o It could also be due to wake rotation as seen in 3D CFD rotor simulations in sheared inflow. It shows that high velocity flow at one side of the rotor is rotated down towards the ground and the opposite on the other side of the turbine.

Response: If it is due to wake rotation, shouldn't the wind speed on the right edge of the deficit be higher than the ambient wind speed from the metmast? Currently, the wind speed agrees with the ambient wind speed.

Line 323:

- … "In this range both turbines operate under optimal and most efficient conditions resulting in maximum energy output from the wind. The thrust coefficient is constant in this region. Therefore, the axial induction and the wind speed deficit normalized by the turbine's inflow wind speed are also expected to be constant for similar ambient conditions over this wind speed range."
    - o Its mentioned ".. expected to be constant". What is actually used in the DWM simulations ?
    - o Further down at line 368 is mentioned : ".. that the axial induction of both turbines is slightly different under partial load conditions." So is the detailed aero loading of each of the two turbines are simulated ?

Response: DWM model simulations for the single turbulence intensity bins and both turbine types are carried out and the same axial induction is applied over the whole wind speed

range. That means, each turbine type is modelled separately and all turbulence intensity bins are simulated. The sentence is rephrased as follows: *"DWM model simulations were carried out for both turbine types, since the axial induction of both turbines is slightly different under partial load conditions. To calculate a mean value of the simulated minimum wind speed and thus allow a comparison with the results in Figure 12, simulations with both turbine types are carried out for each turbulence intensity bin and weighted in accordance with the number of measurement results per turbine listed in Table 2."*.

**8. Comparison between measurements and DWM model simulation**

Line 358:

- "For lower turbulence intensities and higher distances (greater than 3D) there is a relatively large discrepancy between measurements and simulations. A similar observation was made in Larsen et al. (2013)."
    - o This comment was on the model before the recalibration so it should be deleted if pointing to the "DWM-Egmond model"

Response: It is rephrased to: *"A similar observation was made in Larsen et al. (2013) with the model version in Madsen et al. (2010). Aiming at the adjustment of the simulated degradation of the wind speed deficit in Larsen et al. (2013) for cases like the one presented here, the DWM model has been recalibrated..."*
The sentence is not deleted here, because it should be pointed out that the method of recalibration is similar to the one in Larsen et al. (2013).

Line 362

- As concerns the results in Figure 10 and Figure 11 for the DWM-Egmond model they seem not to agree with simulations with our DTU implementation of the DWM model, however with the uncertainty of just assuming a similar turbine operation but without knowing the details of the turbine
    - o The authors are encouraged to share and upload more details of their simulations so that the results can be checked with an original implementation of the so-called DWM-Egmond model.
    - o Further, it is proposed to show a figure with e.g. the mean velocity of the wake deficit or the mean velocity cubed (to show reduction in power of the downstream turbine) and otherwise in the same way as Figure 9. The mean velocity is a more robust characterization of the wake deficit than the minimum value velocity within the deficit. The minimum value can easily be influence by the details of the aerodynamic modelling of the turbine.

Response: A comparison between the static deficit, respectively the solution of the thin-shear layer equations with an implementation of the DTU has already been carried out. The model has been compared to the Python implementation of Jaime Yikon Liew. The two implementations match very well (see following figures). The figures show results from the so-called DWM-Egmond model of both model implementations and their difference ($\varepsilon$ is the mean difference).

[Figure]

The comparison between the two implementations can be found here:
https://github.com/jaimeliew1/dwm_benchmark

The normalized mean wind speed for all turbulence intensity bins are illustrated in the following:

[Figure]

The mean wind speed over a distance of +/- 60m from the wake center is illustrated. Furthermore, a graph from the RMSE between these curves and all model versions is illustrated.

[Figure]

The improvement of the mean wind speed is less clear in comparison to the normalized minimum wind speed. But nevertheless, there is an improvement in almost all turbulence intensity bins or similar good results could be achieved. In the smaller turbulence intensity bins and closer distances, the recalibrated DWM-Keck-c model agrees less well with the measurements. At closer distances the wind speed deficit gets coarse since less scan points are gathered and the influence of the turbulence at the tails is much higher. This leads to an error in the mean wake wind speed but not in the minimum wind speed, which explains these discrepancies. This is the reason why the minimum wake wind speed is illustrated in the paper and used for the recalibration of the DWM-model.

**Some final conclusive remarks**
- There is no discussing of the impact of the findings. Changing the wake recovery characteristics have obviously an impact on power production and loads.
  - For the Egmond aan Zee case the DWM model was as mentioned calibrated to the power reduction of the second turbine in a row relative to the first one for different spacings and turbulence intensities. Using this calibration an overall good correlation of simulated and measured loads was found.
  - Have the present recalibrated model been used for power and load simulations and compared with measurements in the present wind farm?
  - The reviewer finds that due to the above mentioned uncertainties/limitations related to the measurements of the deficits in the meandering frame of reference there will be a bias of the measured deficits being more smooth. Please comment on this view.

Response: The comparison of the recalibrated model with power productions and loads in the wind farm is currently analyzed and will be published soon.
Comments according to the bias in measuring the wind speed deficit in the meandering frame of reference were answered directly at the specific positions above. A graph with DWM model simulations with and without vertical meandering was added.

---

## Referee Report (RR1)

**Final review comments on revised version from March 25 of paper wes-2019-89**

**Measuring dynamic wake characteristics with nacelle mounted LiDAR systems**

by Inga Reinwardt[1], Levin Schilling[1], Peter Dalhoff[1], Dirk Steudel[2], and Michael Breuer[3]

[1]Dep. Mechanical Engineering & Production, HAW Hamburg, Berliner Tor 21, D-20099 Hamburg, Germany

[2]Dep. Turbine Load Calculation, Nordex Energy GmbH, Langenhorner Chaussee 600, D-22419 Hamburg, Germany

[3]Dep. of Fluid Mechanics, Helmut-Schmidt University Hamburg, Holstenhofweg 85, D-22043 Hamburg, Germany

**Comments to revised paper published on March 25, 2020**

The reviewer is satisfied with the response from the author's and the changes in the modified paper published on March 25.

However just a final comment.

The authors write in lines 61-64

*Furthermore, the collected LiDAR measurements are used to recalibrate the DWM model, which enables a more precise modelling of the wake degradation. As a consequence, the calculation of loads and energy yield of the wind farm can be.*

The impact of the recalibration on loads and power is of great importance which would have been valuable to in the same paper as the recalibration. However, it´s mentioned to be ongoing work to be published soon. Maybe in the present paper the authors could write a few lines about what the impact on power and loads qualitatively will be.
A key issue is that in the DWM-Egmond model the calibration of the coupling of the eddy viscosity to ambient turbulence was carried out on basis of turbine power measurements at different spacings and turbulence intensities whereas now it is on basis of wake flow measurements. Apparently, the two calibration methods give different results.

---

## Author Response (AR2)

**Responses to the comment on** *"DWM model calibration using nacelle mounted lidar systems" by Inga Reinwardt et al., manuscript number: wes-2019-89*

**Iteration:** correction

**Comments to the Author:**

Dear authors,

Both referees are satisfied with your revised manuscript and your answers to their questions.

Referee 2 has a last request of modification (see report).

Could you please proceed with this modification and upload again the manuscript?

Best regards

Sandrine Aubrun

The authors write in lines 61-64 :"Furthermore, the collected LiDAR measurements are used to recalibrate the DWM model, which enables a more precise modelling of the wake degradation. As a consequence, the calculation of loads and energy yield of the wind farm can be."

The impact of the recalibration on loads and power is of great importance which would have been valuable to in the same paper as the recalibration. However, it´s mentioned to be ongoing work to be published soon. Maybe in the present paper the authors could write a few lines about what the impact on power and loads qualitatively will be.

A key issue is that in the DWM-Egmond model the calibration of the coupling of the eddy viscosity to ambient turbulence was carried out on basis of turbine power measurements at different spacings and turbulence intensities whereas now it is on basis of wake flow measurements. Apparently, the two calibration methods give different results.

Response:  An example of DWM-Model load and power simulations for different inflow conditions are given in Figures 1 to 9. Simulations with a wind direction of -30° to 30° in 2° steps were carried out, whereby a wind direction of 0° means full wake. The simulations are valid for an ambient wind speed of 8 m/s and a distance between the upstream and downstream turbine of 3.61D (D=117 m). Results of the normalized damage equivalent load (DEL) of the flapwise moment for an ambient turbulence intensity of 4 %, 8 % and 12 % are given in Figures 1 to 3. The corresponding tower bottom fore-aft moment and the power are shown in Figures 4 to 9. The recalibration significantly affects the lower turbulence intensity simulations, especially at partial wake conditions. The influence of the recalibration on the power output is considerably lower than the influence in the flapwise and tower fore-aft loads. This could also be seen in the mean DEL respectively power over all simulated wind directions, which are summarized in Tables 1 to 3. The mean value is taken with respect to the woehler coefficient, which are given in the Figure captions. The difference between the original DWM-Keck model and the recalibrated DWM-Keck-c model regarding loads is about 13 % for a turbulence intensity of 4 %, whereas the difference in power is less than 1 %. Even the power difference between the DWM-Egmond model and the DWM-Keck-c model is only 7 %, which could explain the difference in the calibration results presented here and the calibration of the DWM-Egmond model.

To give a brief overview of the influence of the recalibration into turbine power and loads following phrase has been added in the conclusion of the paper: *"Simulations have shown that the recalibration of the DWM-Keck model can lead up to 13 % lower loads in the turbulent depending components in cases with small turbine distances and low turbulence intensities, whereas for higher turbulence intensities (>12 %) the difference between the original DWM-Keck model and the recalibrated model is less than 5 %. The overall influence of the recalibration on the power output is low (<2 % for all turbulence intensities)."*

*Table 1: Accumulated normalized DELs and power over all inflow conditions for an ambient wind speed of 8 m/s and for an ambient turbulence intensity of 4 %*

| Model | Flapwise DEL [-] | Tower bottom DEL [-] | Power [-] |
|---|---|---|---|
| DWM-Keck | 2.626 | 2.303 | 0.764 |
| DWM-Egmond | 2.851 | 2.583 | 0.726 |
| DWM-Keck-c | 2.292 | 2.015 | 0.775 |

*Table 2: Accumulated normalized DELs and power over all inflow conditions for an ambient wind speed of 8m/s and for an ambient turbulence intensity of 8 %*

| Model | Flapwise DEL [-] | Tower bottom DEL [-] | Power [-] |
|---|---|---|---|
| DWM-Keck | 1.973 | 2.024 | 0.772 |
| DWM-Egmond | 2.175 | 2.305 | 0.735 |
| DWM-Keck-c | 1.816 | 1.840 | 0.784 |

*Table 3: Accumulated normalized DELs and power over all inflow conditions for an ambient wind speed of 8 m/s and for an ambient turbulence intensity of 12 %*

| Model | Flapwise DEL [-] | Tower bottom DEL [-] | Power [-] |
|---|---|---|---|
| DWM-Keck | 1.617 | 1.673 | 0.786 |
| DWM-Egmond | 1.816 | 1.951 | 0.746 |
| DWM-Keck-c | 1.547 | 1.586 | 0.795 |

[Figure]

*Figure 1: Normalized flapwise blade root DEL over different wind directions for an ambient wind speed of 8 m/s, an ambient turbulence intensity of 4% and a downstream distance of 3.61D. The DELs have been calculated with a wöhler coefficient of 10.*

[Figure]

*Figure 2: Normalized flapwise blade root DEL over different wind directions for an ambient wind speed of 8 m/s, an ambient turbulence intensity of 8 % and a downstream distance of 3.61D. The DELs have been calculated with a wöhler coefficient of 10.*

[Figure]

*Figure 3: Normalized flapwise blade root DEL over different wind directions for an ambient wind speed of 8 m/s, an ambient turbulence intensity of 12 % and a downstream distance of 3.61D. The DELs have been calculated with a wöhler coefficient of 10.*

[Figure]

*Figure 4: Normalized tower bottom fore-aft DEL over different wind directions for an ambient wind speed of 8 m/s, an ambient turbulence intensity of 4 % and a downstream distance of 3.61D. The DELs have been calculated with a wöhler coefficient of 4.*

[Figure]

*Figure 5: Normalized tower bottom fore-aft DEL over different wind directions for an ambient wind speed of 8 m/s, an ambient turbulence intensity of 8 % and a downstream distance of 3.61D. The DELs have been calculated with a wöhler coefficient of 4.*

[Figure]

*Figure 6: Normalized tower bottom fore-aft DEL over different wind directions for an ambient wind speed of 8 m/s, an ambient turbulence intensity of 12 % and a downstream distance of 3.61D. The DELs have been calculated with a wöhler coefficient of 4.*

[Figure]

*Figure 7: Normalized power over different wind directions for an ambient wind speed of 8 m/s, an ambient turbulence intensity of 4 % and a downstream distance of 3.61D.*

[Figure]

*Figure 8: Normalized power over different wind directions for an ambient wind speed of 8 m/s, an ambient turbulence intensity of 8 % and a downstream distance of 3.61D.*

[Figure]

*Figure 9: Normalized power over different wind directions for an ambient wind speed of 8 m/s, an ambient turbulence intensity of 12 % and a downstream distance of 3.61D.*